



**New particle formation inside ice clouds: In-situ observations in the tropical**
**tropopause layer of the 2017 Asian Monsoon Anticyclone**
Ralf Weigel[1], Christoph Mahnke[2,6], Manuel Baumgartner[1,3], Martina Krämer[1,4], Peter Spichtinger[1],
Nicole Spelten[4], Armin Afchine[4], Christian Rolf[4], Silvia Viciani[5], Francesco D'Amato[5], Holger
Tost[1], and Stephan Borrmann[1,2]
[1]Institut für Physik der Atmosphäre, Johannes Gutenberg Universität, Mainz, Germany
[2]Partikelchemie, Max-Planck-Institut für Chemie, Mainz, Germany
[3]Zentrum für Datenverarbeitung, Johannes Gutenberg University, Mainz, Germany
[4]Institute of Energy and Climate Research (IEK-7), Forschungszentrum Jülich, Jülich, Germany
[5]National Institute of Optics - National Research Council (CNR-INO), Florence, Italy
[6]now at the Institute of Energy and Climate Research (IEK-8), Forschungszentrum Jülich, Jülich,
Germany
**Abstract**
From 27 July to 10 August 2017 the airborne StratoClim mission took place in Kathmandu, Nepal
where eight mission flights were conducted with the M-55 *Geophysica* up to altitudes of 20 km.
New Particle Formation (NPF) was identified by the abundant presence of ultrafine aerosols,
with particle diameters $d_p$ smaller than 15 nm, which were *in-situ* detected by means of
condensation nuclei counting techniques. NPF fields in clear-skies as well as in the presence of
cloud ice particles ($d_p > 3\,\mu m$) were encountered at upper troposphere / lowermost
stratosphere (UT/LS) levels and within the Asian Monsoon Anticyclone (AMA). NPF-generated
ultrafine particles in elevated concentrations ($N_{uf}$) were frequently found together with cloud ice
(in number concentrations $N_{ice}$ of up to 3 cm⁻³) at heights between ~ 11 km and 16 km. From a
total measurement time of ~ 22.5 hours above 10 km altitude, in-cloud NPF was in sum detected
over ~ 1.3 hours (~ 50 % of all NPF records throughout StratoClim). Maximum $N_{uf}$ of up to
~ 11000 cm⁻³ were detected coincidently with intermediate ice particle concentrations $N_{ice}$ of
0.05 – 0.1 cm⁻³ at comparatively moderate carbon monoxide (CO) contents of ~ 90 -
100 nmol mol⁻¹. Neither under clear-sky nor during in-cloud NPF do the highest $N_{uf}$
concentrations correlate with the highest CO mixing ratios, suggesting that an elevated pollutant
load is not a prerequisite for NPF. Under clear-air conditions, NPF with elevated $N_{uf}$ (> 8000 cm⁻³) occurred slightly less often than within clouds. In the presence of cloud ice, NPF with $N_{uf}$
between 1500 – 4000 cm⁻³ were observed about twice as often as under clear air conditions.
When ice water contents exceeded 1000 µmol mol⁻¹ in very cold air (< 195 K) at tropopause
levels NPF was not found. This may indicate a reduction of NPF once a strong overshoot is
prevalent together with the presence of mainly *liquid-origin* ice particles. In the presence of *in-
situ* cirrus near the cold point tropopause very recent NPF or events of remarkable strength
(mixing ratios $n_{uf} > 5000\,mg⁻¹$) were rarely observed (~ 6 % of in-cloud NPF data). For
specifying the constraining mechanisms for NPF possibly imposed by the microphysical





properties of the cloud elements, the integral radius ($IR$) of the ice cloud population was
identified as the most practicable indicator. Neither of both, the number of ice particles or the
free distance between the ice particles, is clearly related to the NPF-rate detected. The results of
a numerical simulation indicates how the $IR$ affects the supersaturation of a condensable vapour,
such as sulphuric acid, and that $IR$ determines the effective limitation of NPF rates due to cloud
ice.

## 1.  Introduction

The process of gas-to-particle conversion, also denoted as homogeneous aerosol nucleation and
most commonly known as new particle formation (NPF), constitutes one of the most effective
sources of atmospheric aerosols and cloud condensation nuclei, which could promote the cloud
formation at intermediate and upper tropospheric altitudes (e.g. Spracklen et al. (2006);
Merikanto et al. (2009); Dunne et al. (2016); Gordon et al. (2017)). Sulphuric acid ($H_2SO_4$) and
water ($H_2O$) presumably are important chemical compounds involved in the NPF process which,
moreover, is likely aided when ions come into play at elevated altitudes and cold temperatures
within the atmosphere (Lee et al. (2003); Kazil et al. (2008); Weigel et al. (2011); Duplissy et al.
(2016)). It was suggested that a ternary nucleation process involves, apart from sulphuric acid
and water, an additional constituent such as ammonia ($NH_3$; Ball et al. (1999); Benson et al.
(2009); Höpfner et al. (2019)). Experimental studies at the CLOUD (Cosmics Leaving OUtdoor
Droplets) chamber confirmed that NPF rates are substantially elevated within this ternary
$H_2SO_4$-$H_2O$-$NH_3$ System (e.g. Kirkby et al. (2011); Kürten et al. (2016); Kürten (2019)).
From the CLOUD experiments, which were performed under a variety of controlled conditions, it
can be deduced that the intensity of NPF (the formation rate of new particles per air volume and
per time unit) depends on the concentration of the NPF precursors. The results of individual
experiments (Kürten et al. (2015); Kürten et al. (2016)) at different and elevated concentrations
of the $H_2SO_4$ solution, always at supersaturated states, show that the nucleation rates are
strongly associated with the precursor concentrations. Temperature determines the degree of
supersaturation, which implies that even high precursor concentrations may result in a weak
NPF rate, and vice versa. In particular, for ternary or multi-component NPF, the degree of
supersaturation as a function of temperature remains indeterminable. Therefore, the chamber
experiments allow for studying the nucleation rate as a function of the precursor concentration
at different temperatures, i.e. at supersaturation ratios, which are specific, but mostly unknown,
with respect to the system of nucleating substances (involving $H_2SO_4$, $H_2O$, and $NH_3$). The
complexity increases with sulphuric acid nucleation systems involving besides $NH_3$ also nitric
acid ($HNO_3$) (Wang et al., 2020) or oxidised organic vapours (Riccobono et al., 2014), all of which
may promote the NPF process at supersaturations lower than required for pure $H_2SO_4$ solutions.





The role of organic substances in connection with NPF could be of particular importance in the
tropical UT/LS as has been indicated by (Schulz et al., 2018) and (Andreae et al., 2018). The time
series of a nucleation event within the CLOUD chamber (supplementary material of Kirkby et al.
(2011)) shows, however, that the nucleation rate remains elevated as long as the amount of
precursors is kept at a constant level as investigated by means of the CLOUD experiments. Under
real conditions in the atmosphere, however, the concentration of precursor material is spatially
and temporally highly variable. Temperature fluctuations affect the degree of precursor
supersaturation; hence, even low precursor concentrations may result in elevated
supersaturations and intense NPF. The influence of third or multiple substances possibly
involved in the NPF process is not unambiguously detectable or even quantifiable in the
ultrafine particle population due to the current lack of instrumentation capable of analysing the
chemical composition of such small particles directly.
By means of ground based as well as airborne *in-situ* measurements, NPF was frequently
observed to occur at various conditions and atmospheric altitudes (Kerminen et al., 2018).
Recently, Williamson et al. (2019) compiled a comprehensive data set of *in-situ* NPF
observations at altitudes from 180 m above sea level to up to ~ 12 km, thereby covering a
latitude range from 80° North to 70° South alongside the Americas, and by probing air over both
oceans, the Pacific and the Atlantic. In tropical regions, most of the *in-situ* NPF observations were
made below the level of zero net radiative heating, i.e. at altitudes where subsidence or cloud
formation is still well capable to efficiently remove or scavenge aerosol particles.
Investigations concerning the occurrence of NPF within clouds, or in their immediate vicinity,
are sparse and are mainly limited to tropospheric altitudes. The region above tropospheric
clouds seems favourable for NPF to occur, and possible reasons for this are discussed by Wehner
et al. (2015). Furthermore, NPF was found to be an important process inside the convective
outflows (e.g. Twohy et al. (2002); Waddicor et al. (2012)). From measurements in the upper
troposphere it is commonly assumed that the occurrence of NPF is directly connected to deep
convective cloud systems (e.g. de Reus et al. (2001); Clarke and Kapustin (2002); Weigelt et al.
(2009); Andreae et al. (2018)). The relationship between NPF and ice clouds is discussed in this
study, whilst the immediate connection of NPF and deep convective events is addressed in
Weigel et al. (2020a).
During *in-situ* measurements aboard the NASA high altitude research aircraft WB-57, Lee et al.
(2004) observed nucleation events inside subtropical and tropical cirrus clouds between 7 and
16 km over Florida. The same authors summarise that they found recent occurrence of NPF in
72 % of their measurements within clouds. Despite the conceptual notion that the presence of
cloud elements generally inhibits the formation of new particles, Kazil et al. (2007)





demonstrated by means of model simulations that new sulphate aerosol can form within ice
clouds such as cirrus. New particles are also produced in the anvil region and cirrus decks of
Mesoscale Convective Systems (MCS) over West Africa (Frey et al., 2011). The particular role of
mid-latitude MCS as a source of freshly formed aerosol within the upper troposphere was
already suggested by Twohy et al. (2002), based on the detection of increased concentrations of
particles with size diameter ($d_p$) greater than 25 nm, concurrently with elevated particle
volatility. In the region of the Tropical Transition Layer (TTL) over South America, Australia and
West Africa, the *in-situ* measurements by Weigel et al. (2011) revealed nucleation mode
particles in elevated number concentrations likely resulting from recent NPF. Based on
coincident detections of abundant nucleation mode particles together with cloud elements (i.e.
ice particles of diameters 2.7 μm $< d_p <$ 1.6 mm) in ice number concentrations always below
$\sim 2$ cm$^{-3}$ the authors concluded that the occurrence of NPF is mainly limited by the number of
cloud particles. The underlying concept is that the surfaces of the cloud elements either
scavenge the NPF-produced aerosol particles or remove the nucleating vapour molecules prior
to the NPF process.
Regarding the occurrence of NPF in conjunction with the presence of upper tropospheric ice
clouds, still several unspecified details remain:

1) what are the sets of chemical species acting as NPF precursor,

2) does NPF possibly require (or not) contributions by cosmic radiation, ions (Lovejoy et al.

(2004); Kazil et al. (2008); Weigel et al. (2011)) or chemical agents or catalysts (e.g.

Kürten (2019)),

3) which are the advantageous thermodynamic conditions for NPF within a cloud, and

4) the conditions under which NPF is suppressed by the presence of ice particles of certain

size and/or number.

Comprehensive understanding of these processes and their influences under real atmospheric
conditions potentially contributes to narrow down the cloud type and the in-cloud location
where NPF preferentially occurs, in order to obtain estimates (in particular for modelling
purposes) concerning the importance of in-cloud NPF. Furthermore, the question could arise
how the ultrafine particles generated by in-cloud NPF are processed: for example, if the ultrafine
particles disperse as contribution to the clear air background aerosol as soon as the cloud
elements evaporate, or if, in persistent clouds, the ultrafine particles are captured by present ice
particles. In the context of the Asian Monsoon Anticyclone (AMA) it is important to clarify the
origin of observed enhancements of aerosols in the embedded Asian Tropopause Aerosol Layer
(ATAL, cf. Vernier et al. (2011); Vernier et al. (2018)). NPF could well be an important source of
aerosol particles which are then available for further processing to form the ATAL (Höpfner et





al. (2019); He et al. (2019); Mahnke et al. (2020)). Furthermore, the relative contribution of in-
cloud versus clear-air NPF is of importance in this context.
The Asian Monsoon Anticyclone (AMA) constitutes one of the most important weather systems,
which mainly determines the circulation in the Upper Troposphere/Lower Stratosphere
(UT/LS) during monsoon season over the Indian subcontinent. The AMA is usually associated
with extensive deep convection capable of transporting polluted air from the regional boundary
layer (BL) to high altitudes. From the beginning of June through about the end of September, the
large-scale anticyclone persists in the altitude level from the UT reaching up into LS regions (e.g.
Randel and Park (2006); Park et al. (2007)) spanning over longitudes from East Asia to the
Middle East/East Africa (e.g. Vogel et al. (2014); Vogel et al. (2019)). The system's anticyclonic
rotation induces the development of a horizontal transport barrier within in the UT/LS (Ploeger
et al., 2015) reducing isentropic exchange between the interior of the AMA and the anticyclone's
surroundings. The vertical upward transport within the Asian monsoon circulation is
understood as an effective pathway for young air of BL - origin (Vogel et al., 2019) to rapidly
reach UT/LS altitudes, accompanied by various constituents such as pollutants and further
gaseous material (Pan et al., 2016) and in particular water vapour (Ploeger et al., 2013). To
which extent the stratospheric entry of $H_2O$ is supported by cirrus cloud particles (as a result of
overshooting convection or ice formation due to local dynamics; de Reus et al. (2009); Corti et al.
(2008)) is currently under debate (Ueyama et al. (2018), and references therein) and one of the
subjects of a recent study by Krämer et al. (2020). Based on satellite investigations the existence
of the ATAL was explored at tropopause altitudes within the AMA region (Vernier et al. (2011);
Thomason and Vernier (2013)). Therefore, the constituents of the uplifted young air from low
altitudes may also comprise the precursor material from anthropogenic (Vernier et al. (2015);
Yu et al. (2015); Höpfner et al. (2019); Mahnke et al. (2020)) and other sources to develop and
maintain the observed aerosol layer, most likely due to NPF occurring at levels between
approximately 14 km and the tropopause.
This study reports on the frequent occurrence of NPF in the presence of cloud ice in the
tropopause region over the Indian subcontinent during the Asian monsoon season. All
measurement data shown herein were acquired during StratoClim 2017 (in July/August 2017)
based at Kathmandu, Nepal, and conducted with the M-55 *Geophysica* that operates up to 20 km
altitude. NPF was observed with almost equivalent extent in clear-air as well as, under certain
conditions, in the midst of cloud ice particles. This investigation aims at summarising the various
conditions under which NPF was observed coincidently with cloud ice particles. Close
examination of the measured data revealed that potential artefacts on the aerosol
measurements due to the presence of ice particles, as suggested by Williamson et al. (2019), are





largely excludable for the StratoClim data set (cf. Appendix A). The caveats limiting the
magnitude of encountered NPF are examined, as are the possibly constraining mechanisms
imposed by the cloud elements' microphysical properties. The frequency of NPF observation in
coincidence with elevated ice particle densities puts emphasis on the importance of the
tropopause region within the AMA as an effective source region of freshly nucleated aerosol.

## 183    2    The StratoClim field campaign in 2017, instruments and methods

During the Asian monsoon season, between 27 July and 10 August 2017, a total of eight scientific
flights was conducted above parts of the Indian subcontinent, out of Kathmandu, Nepal
(27° 42' 3'' N, 85° 21' 42'' E) throughout StratoClim 2017 (cf. Figure 1). Some of these flights
partly led out of the Nepalese airspace, to east India, Bangladesh and the farthest north of the
Bay of Bengal. The occurrence of NPF was encountered (cf. Figure 1) during each flight, either in
clear air or in the presence of cloud (ice) particles.

## 190    2.1 Number concentration of sub-micrometre sized particles

The 4-channel continuous flow condensation particle counter COPAS (COndensation PArticle
counting System; Weigel et al. (2009)), which was operated with the chlorofluorocarbon (FC 43)
as working liquid, was used for measuring aerosol particle number concentrations. Particle
detection and data storage occurred at 1-Hz frequency. For the reduction of the statistical noise
in the raw signals, which are recorded directly from the scattered light detectors of the COPAS
instrument, the 1-Hz raw data are pre-processed by a 15-second running average. The COPAS
channels were set to different 50 %-detection particle diameters $d_{p50}$ (i.e. 6 nm, 10 nm, and
15 nm). By counting aerosols (with $d_{p50} = 10$ nm) downstream of a heated ($\sim 270°C$) sample
flow line, a fourth COPAS channel measured particle concentrations of non-volatile (nv) or
refractory particles (e.g. soot, mineral dust, metallic aerosol material as well as, e.g., organic
material mixtures not evaporating at 270 °C, etc. ). By means of laboratory experiments, the
aerosol heating device was demonstrated to vaporise more than 98 % of $H_2SO_4$-$H_2O$ particles at
pressures between 70 and 300 hPa (Weigel et al., 2009). The aerosol sampling occurred via the
forward facing aerosol inlet of COPAS, a custom-made reproduction of the inlet used on board
the NASA ER-2 (Wilson et al., 1992). The use of this aerosol inlet also facilitated Lee et al. (2004)
to measure in-cloud NPF. The inlet allows for aerosol sampling well outside the boundary layer
of the aircraft. The inlet's geometry comprises two serial diffusors to sample air (at super-
isokinetic conditions) with decelerated flow speed compared to the ambient free flow. The
largest particle diameter that is detectable with the COPAS system is confined by the inlet
geometry, and it is estimated that sub-micrometre sized particles enter the aerosol inlet and
pass the aerosol lines without significant particle losses (Weigel et al., 2009). However, aerosol
particles with diameter of up to 5 µm may occasionally be aspired by the COPAS inlet, but are





clearly undersampled (Ebert et al., 2016). The COPAS measurement uncertainty is about 15 %
for stratospheric particle concentrations, mainly due to uncertainties in the volume flow and as a
result from particle counting statistics. The COPAS is an established instrument for high altitude
application, its performance was characterised by Weigel et al. (2009) and COPAS data were
used and discussed in various studies (e.g. in Curtius et al. (2005); Borrmann et al. (2010); Frey
et al. (2011); Weigel et al. (2011); Weigel et al. (2014); Schumann et al. (2017); Höpfner et al.

(2019)).

**2.2 Terminology and notations**
The measured particle number concentrations $N$ are provided in units of particle number per
cubic centimetre of sampled air (ambient conditions). To compare aerosol observations from
different pressure altitudes and, e.g., for correlations with mixing ratios of trace gases, COPAS
measurements are also given as mixing ratio $n$ in units of particles per milligram of air (mg$^{-1}$) as
calculated based on the 1-Hz-resolved data of ambient static pressure and temperature (cf.
Section 3.5). Hereafter, $n_{10}$ denotes the mixing ratio of particles with diameters larger than
10 nm. The detection of the particle number concentration or mixing ratio at different $d_{p50}$ (i.e.
$N_6$ or $n_6$ for particles with $d_p > 6$ nm, and $N_{15}$ or $n_{15}$ for $d_p > 15$ nm), aims at the identification of
recent new particle formation, principally based on the difference of both quantities (cf. Weigel
et al. (2011)). The number concentration of ultrafine aerosol particles (hereafter denoted as $N_{uf}$)
is calculated from the difference $N_6 - N_{15} = N_{6-15}$ and serves as an indication for recent NPF if the
designated NPF criterion is met:
$$0.8 \cdot N_6 - 1.2 \cdot N_{15} > 0 \qquad (1),$$

based on the principle definition used by Weigel et al. (2011). Further details concerning the
corrections applied to the measured COPAS data, which were obtained throughout
StratoClim 2017, are provided by Weigel et al (2020a), where the empirical parameters of 0.8
and 1.2 are introduced.
If compliant with the NPF criterion, a series of data points is a designated event if measured
number concentrations (or mixing ratios) of ultrafine particles continuously remain greater than
zero over at least five consecutive seconds. Caveats with this event definition are inherent for
observations as short as 1 – 5 seconds. Due to the detector's signal-to-noise-ratio and counting
statistics, the given quantity and durations of too short events bear significant uncertainties. In
addition, however, this event definition may prevent resolving very fine spatial structures (i.e.
horizontally on 150 m-scales, vertically on 10 m-scales) of NPF fields. With the mean airspeed of
the M-55 *Geophysica* ($\sim 154 \pm 39$ m s$^{-1}$), the event definition implies that within five seconds a
horizontal distance of $\sim 770$ m (in flight direction) is covered. The total of 308 individual





detections of elevated $N_{uf}$ coincidently with the presence of cloud elements, 104 of which
fulfilled the event criterion. Note that the in-cloud NPF events discussed herein are partially
embedded in a larger NPF fields, which are identified by successive, uninterrupted detections of
elevated $N_{uf}$. Within this larger NPF fields the duration of simultaneous ice particles detection
could be shorter or interrupted. One or more in-cloud NPF events could be subsets of NPF
events with continuously elevated $N_{uf}$ concentration, which are generally discussed by Weigel et
al. (2020a). They (*ibid.*) also present further details concerning the duration of NPF events, the
persistence of the freshly formed particles in the ultrafine size range, and the presence of non-
volatile particles under NPF conditions during StratoClim 2017.
The NPF-rate and, hence, the intensity of NPF varies with the degree of supersaturation of the
NPF precursor (Kirkby et al. (2011), Kürten et al. (2016)). Hereafter, the strength of a NPF event
is classified as
(1)  *excessive* NPF if detected aerosol densities of ultrafine particles exceed

• mixing ratios of 10000 mg$^{-1}$ or

• number concentrations of 5000 cm$^{-3}$,

(2)  *intermediate* NPF when number densities of ultrafine particles range at

• mixing ratios of 1000 mg$^{-1}$ < $n_{uf}$ < 10000 mg$^{-1}$ or

• number concentrations of 500 cm$^{-3}$ < $N_{uf}$ < 5000 cm$^{-3}$, and

(3)  *weak* NPF when

• mixing ratios $n_{uf}$ remain below  1000 mg$^{-1}$, or

• number concentrations $N_{uf}$ of less than 500 cm$^{-3}$ are detected.

(4)  The notation *most intense* NPF is often used synonymously with *most recent* NPF in the

following.

As the persistence of the particles in the ultrafine size range is short (i.e. a few hours only, cf.
Weigel et al. 2020a), an intense NPF event could still be in process when observed, or it had been
completed very recently, i.e. 1-2 hours prior to the detection. However, for a NPF encounter with
low or intermediate number densities of ultrafine particles ($n_{uf}$ or $N_{uf}$), the conclusions
concerning the event's age remain ambiguous, as it can be caused by weakly proceeding NPF or
by NPF, which had occurred several hours ago.
**2.3 Cloud particle and water vapour detection**
The NIXE-CAPS (New Ice eXpEriment: Cloud and Aerosol Particle Spectrometer, in the following
denoted as NIXE) was deployed during StratoClim for measuring the number size distribution in
the cloud particles' diameter size range of 3 – 930 µm with 1-Hz resolution (Luebke et al. (2016);
Costa et al. (2017)). Comprehensive numerical analyses by means of computations fluid



dynamics (CFD) were carried out by Afchine et al. (2018) to investigate the impact of the
instrument's position underneath the aircraft wing on the cloud particle detection. The NIXE-
CAPS consist of two detectors: the NIXE-CAS-DPOL (Cloud and Aerosol Spectrometer with
Detection of POLarization) and the NIXE-CIPg (Cloud Imaging Probe – grayscale). Compiling
measured data of both independent detectors delivers microphysical properties, in terms of size
and number, of particles with diameters ranging from 0.61 μm to 937 μm. The methods of post
flight data processing and corrections were described by Luebke et al. (2016). Cloud particle
detections were recognised as such when particles of diameters > 3 μm were encountered in
numbers greater than zero. Hereafter, the number concentration of ice particles is denoted as
$N_{ice}$ (i.e. $N_{3-937\mu m}$, the number concentration of ice particles with diameters of
3 μm < $d_p$ < 937 μm). The data of ice water content ($IWC$) used herein were ascertained by using
the relationship of cloud particles' mass ($m_p$) to diameter ($d_p$) (Krämer et al. (2016); Luebke et
al. (2016); Afchine et al. (2018)).
The closed-path Lyman-α photo-fragment fluorescence hygrometer FISH (Fast In situ
Stratospheric Hygrometer; cf. Zöger et al. (1999) and Meyer et al. (2015)) allows for 1-Hz
resolved measurements of the atmosphere's gaseous and solid phase water, denoted as total
water or $H_2O_{tot}$. The FISH detection of $H_2O_{tot}$ covers mixing ratios of 1 - 1000 μmol mol$^{-1}$ over
atmospheric pressures ranging from 50 hPa to 500 hPa with an accuracy and precision of 6 –
8 % and 0.3 μmol mol$^{-1}$. The ice water content ($IWC$) was calculated by subtracting the $H_2O_{Gas}$
(measured by another Lyman-α detector, FLASH, the FLuorescent Airborne Stratospheric
Hygrometer) from $H_2O_{tot}$. For further detail concerning the data processing, see Afchine et al.
(2018). Dependent on ambient temperatures, the smallest $IWC$ detectable by the FISH
instrument is about $1 \cdot 10^{-3}$ μmol mol$^{-1}$ and $20 \cdot 10^{-3}$ μmol mol$^{-1}$, which corresponds to
approximately $1 – 20 \cdot 10^{-4}$ mg m$^{-3}$ (Afchine et al., 2018).
To cover the wide range of $IWC$ observed during the StratoClim mission (from thousandths to
thousands of μmol mol$^{-1}$) the complementary data sets of NIXE-CAPS and FISH concerning $IWC$
were merged. Thus, when large ice particles were abundant, causing $IWC$ of hundreds to
thousands of μmol mol$^{-1}$, mostly NIXE-CAPS data contributed to the resulting $IWC$ data. In
contrast, low numbers of small ice particles caused the FISH instrument to provide the most
reliable $IWC$. The overall uncertainty of given $IWC$ values were estimated to be ~ 20 % (Krämer
et al., 2020).
Based on cloud particle (NIXE-CAPS) and water vapour (FISH) measurements, ice particle
shattering, which could be indicated e.g. by short bursts of small ice particles, remained
unobserved throughout StratoClim (Krämer et al., 2020).





### 2.4 Carbon monoxide


In the troposphere, carbon monoxide (CO) is a component of atmospheric pollution (Park et al.,
2009), the main sources of which are both natural and anthropogenic (including combustion,
and the oxidation of hydrocarbons). It is assumed that the contributions to the tropospheric CO
budget almost equivalently originate from: (1) its photochemical production and (2) directly
from sources located at the surface. Mainly the oxidation with hydroxyl radical (OH) depletes CO
within the atmosphere (Logan et al. (1981); Yin et al. (2015)). CO mixing ratios are well suitable
and often used as tropospheric tracer for air's transport (a) within the troposphere, (b) across
the tropopause, and (c) within the lower stratosphere. In the free troposphere, CO mixing ratios
range from unpolluted 50 nmol mol$^{-1}$ up to mixing ratios well exceeding 700 nmol mol$^{-1}$ in close
vicinity to emission sources (Clerbaux et al. (2008), Park et al. (2009)). Inside the AMA and up to
15 km altitude, CO mixing ratios remain comparatively high ($\gtrsim$ 100 nmol mol$^{-1}$), while between
15 km and 20 km altitude the CO mixing ratios decrease monotonically down to $\sim$ 40 nmol mol$^{-1}$
(Park et al., 2009).
During the StratoClim mission, the mixing ratio of CO was measured by means of the Tunable
Diode Laser (TDL) technique implied in the revised version of the Cryogenically Operated Laser
Diode (COLD) spectrometer. Compared to the previous instrument version (4 s temporal
resolution, Viciani et al. (2008)), COLD-2 integrates improvements (Viciani et al., 2018)
regarding:
1) an increased measurement's resolution by a factor of four,
2) an enhanced in-flight sensitivity of the COLD-2 spectrometer (ranging at $\sim$ 2 nmol mol$^{-1}$
at integration times of 1 s) , and
3) an accuracy of 3 % is specified for the CO measurement with COLD-2.
Within the data set of simultaneous measurements with COPAS, COLD-2 detected minimum and
maximum CO mixing ratios of 14 nmol mol$^{-1}$ and 153 nmol mol$^{-1}$, respectively.

### 2.5 Data of ambient temperature and static pressure


The atmospheric temperature and pressure data were taken from the Unit for Connection with
the Scientific Equipment (UCSE, Sokolov and Lepuchov (1998)), a part of the navigational
system of the M-55 *Geophysica*. UCSE data are provided as 1-Hz-resolved ambient pressure
(with an accuracy of ±1 hPa) and temperature (±2 K accuracy).
The potential temperature $\theta$ is calculated correspondingly with 1-Hz resolution in compliance
with the definition by the World Meteorological Organization (WMO, 1966). Note that for the
given vertical temperature gradients and over the $\theta$-range covered during StratoClim 2017 (i.e.



up to ~ 477 K), the WMO recommended calculation of $\theta$ differs only by up to ~ 1 K from the
values obtained by using the recently reappraised $\theta$-calculation (Baumgartner et al., 2020).

## 3  Observations and results

During StratoClim 2017, eight mission flights were conducted with a total of 36.6 hours, whereas
over a total of 6.42 hours ice clouds were encountered at air temperatures colder than 240 K.
The cirrus cloud observations are described and comprehensively discussed by Krämer et al.
(2020), and thus are only briefly summarised herein. Most of the in-cloud measurements during
StratoClim 2017 were performed at temperatures $\lesssim$ 205 K, corresponding to potential
temperatures above ~ 355 K and altitudes higher than ~ 14 km, i.e. well within the TTL region.
The clouds observed during the Asian monsoon season include: 1) *in-situ* cirrus, which had
formed in calm dynamic situations associated with very slow updraught as well as 2) *liquid-*
*origin* cirrus, the formation of which is connected to overshooting deep convection with elevated
uplift velocities (see Section 5.2), including ice clouds (e.g. anvils) associated with convective
outflow.
At temperatures colder than 205 K, $N_{ice}$ and *IWC* often reached values above their respective
median of 0.031 cm$^{-3}$ (blue dashed line in Figure 2) and ~ 0.2 – 2 µmol mol$^{-1}$ (cf. Figure 6). The
highest observed values at these temperatures are reached with *IWC* of up to 1000 µmol mol$^{-1}$
and a maximum $N_{ice}$ as high as 30 cm$^{-3}$. Moreover, the ice crystals sizes (not shown here) exceed
their corresponding median, hence, comparatively large ice crystals were found up to and
around the cold point tropopause. Such large particles were detected during flights in strong
convection.

### 3.1 The distribution of NPF and the presence of cloud ice particles over day time

During a total of ~ 22.5 hours of COPAS measurement time at altitudes above 10 km ($\theta \gtrsim$ 350 K),
in general, a duration of about 2 hours and 38 minutes was spent under NPF conditions in the
TTL region (~ 11-17.5 km, ~ 355 – 400 K, cf. Weigel et al 2020a). Throughout the entire
StratoClim mission, elevated number densities of ultrafine particles were observed coincidently
with cloud particles ($N_{ice}$ > 0 cm$^{-3}$) over a total of about 1 hour and 17 minutes (cf. Table 1). The
encountered in-cloud NPF events at altitudes between approximately 11 km and 16.5 km
(~ 355 – 385 K) had a mean event duration of 14.5 seconds (ranging from one second to a
maximum of about 300 seconds).
In Figure 2, all NPF detections throughout the StratoClim mission are compiled within a 1-day
time series to illustrate the diurnal variability of the observations. The scale of the time series is
limited to the daytime as the eight mission flights were conducted between 03:30 (UTC) and
12:30 (UTC), which corresponds to local times of 09:15 LT to 18:15 LT. Kathmandu local noon



time corresponds to 22500 seconds of day, or 06:15 UTC, and is tagged with an orange line in
Figure 2. The encounter of NPF is considered as clear-air observation (black data points in
Figure 2, complete StratoClim data set) when simultaneously detected cloud (ice) particle
number concentration $N_{ice}$ remained at 0 cm$^{-3}$. Coincident observations of NPF and cloud (ice)
particles ($N_{ice} > 0$ cm$^{-3}$) are highlighted by red data points in Figure 2. The increased frequency of
abundant cloud ice in the local afternoon may temporally coincide with the typically elevated
convective activity during the second half of a day. However, a temporal dependency was not
observed for the occurrence, the strength or the frequency of NPF. Furthermore, there is no
obvious indication that the number of ice particles present had a direct influence on the NPF
strength. The impression arises that even intense NPF happens almost unaffected only by the
number of present cloud ice particles, as otherwise likely larger differences should be visible
between the $N_{uf}$ maxima in clear air and under in-cloud conditions.

### 394     3.2 Vertical distribution of ultrafine particles in presence/absence of cloud ice
### 395        particles

Figure 3 displays the vertical distribution of NPF-generated ultrafine aerosols in terms of the
mixing ratio $n_{uf}$ as a function of potential temperature. The panel a) of Figure 3 depicts the clear-
air observations of elevated $n_{uf}$ (black) together with those when coincidently ice particles were
detected (red). The coincident observation of ice particles and ultrafine aerosol is vertically
limited to a range of potential temperatures from 355 K to 385 K (cf. also Table 1). Recent
convective overshooting to altitudes above the mean tropopause height (∼ 380 K, averaged over
the period and operation area of StratoClim 2017) is indicated by $\theta$ values significantly larger
than 380 K. Clear-air NPF was sampled also at higher altitudes, i.e. at potential temperatures
above 385 K and up to ∼ 400 K, or at lower altitudes, below 355 K. As indicated by the time
series shown in Figure 2, also the vertical profiles suggest that the strength of NPF was largely
independent from the presence of cloud elements. The data in Figure 3b show that in-cloud NPF
observations were made during each of the eight mission flights (cf. Figure 1). Apparently, in-
cloud NPF is a common phenomenon in connection with the AMA and in presence of prevailing
large convective cloud systems over the Himalayan foothills. The separation of in-cloud and
clear-air conditions of NPF observation is illustrated with the intermediate panels (c and d) of
Figure 3. In this way, the in-cloud NPF observations (c), which occurred in the altitude interval
of ∼ 355 < Θ < 385 K, are opposed to NPF observations that were exclusively made under clear-
air conditions (d) over a vertical range between 355 K and 400 K.
The relationship between CO mixing ratios and NPF occurrence in the tropical UT/LS over West-
Africa (Weigel et al., 2011) indicated a relationship between NPF and ground sources of gaseous
NPF precursor substances (mainly sulphur compounds, potentially also organics) that are





efficiently lifted into the TTL region via convective transport, and not entirely removed by
scavenging. Tost et al. (2010) revealed a substantial underestimation of simulated $SO_2$ compared
to flight observations throughout the SCOUT-O3 mission in 2005. These authors utilised global
chemistry climate model simulations independent of the representation of deep convection, the
results of which indicated that the scavenging of $SO_2$ is weaker than expected and that the
impact of retention is not negligible (*ibid.*). However, a substantial fraction of the well soluble
sulphur dioxide ($SO_2$) may not reach UT/LS altitudes via convective events. Cloud-resolving
numerical modelling revealed a fraction of 40-90 % of $SO_2$, that is capable of reaching the deep
convection's outflow region (Barth et al., 2001), largely consistent with the estimates by Crutzen
and Lawrence (2000). The results of other model studies (Ekman et al., 2006) showed that only
30 % of $SO_2$ from the boundary layer reach cloud top levels. Experimental studies by Jost et al.
(2017) found moderate retention coefficients (0.2 – 0.5) of $SO_2$ in the ice phase of clouds, while
hydrochloric acid (HCl) and nitric acid ($HNO_3$) are entirely retained under ice cloud conditions
(*ibid.*). Once the cloud particles freeze, large fractions of the in-cloud dissolved $SO_2$ could leave
the cloud-ice composite. The $SO_2$ remaining in the cloud ice composite, is released as soon as the
ice sublimates in the region of convective outflow, or underneath, while the ice particles
sediment.
NPF should most frequently occur in air enriched with precursor material and which
experienced rapid vertical uplift. An indicator for the air masses' pollutant load and/or contact
with the boundary layer and recent vertical uplift is provided by air's carbon monoxide (CO)
content. According to the CO mixing ratio when $n_{uf}$ mixing ratios were elevated (colour coding in
panels e and f of Figure 3), none of the different NPF conditions, in clear air or in the presence of
ice particles, shows the highest number of ultrafine particles together with highest CO mixing
ratio. In fact, the highest densities of ultrafine particles were observed at comparatively
moderate CO mixing ratios of ∼ 90 - 100 nmol mol$^{-1}$. This largely agrees with the in-situ
measurements of correlations between CO mixing ratio and NPF obtained at similar altitudes in
the region of Mesoscale Convective Systems during the West African Monsoon (Frey et al.
(2011); Weigel et al. (2011)) although based on a smaller data set of coincident CO and particle
detections. Moderately high or intermediate CO mixing ratios may result from the dilution of
young, CO-enriched air with aged and processed air masses, which would reduce both the CO
content of the air and its NPF precursor concentration. As soon as the thermodynamic
conditions for NPF are reached during transport, the formation process may be initialised, while
the content of diluted CO could indicate an unremarkable pollution state of the probed air. In air
masses with lowest CO content (∼ 40 - 60 nmol mol$^{-1}$) NPF was observed only above the
tropopause ($\theta$ > 380 K) and in the absence of ice particles with $n_{uf}$ ranging from 300 mg$^{-1}$ to a
maximum of 2000 mg$^{-1}$ (Weigel et al., 2020a).





Most intense NPF, i.e. with highest densities of ultrafine aerosol particles, were found below the
tropopause ($\sim$ 380 K). In the presence of ice particles (as in clear air), elevated $n_{uf}$ values were
also encountered at low CO mixing ratios, below $\sim$ 70 nmol mol$^{-1}$, at potential temperatures of
370 - 380 K. Under clear-air conditions, NPF occurs at much lower CO mixing ratios, which is
shown by the $n_{uf}$ vertical profile (Figure 3f). The results of Figure 3 indicate that NPF
predominantly occurred in an altitude band between 350 K and 380 K (corresponding to $\sim$ 8.5 –
16.5 km) with $n_{uf}$ in the range of about 1000 to 50000 mg$^{-1}$. The $n_{uf}$ values of NPF in ice clouds
do generally not differ from those of NPF under clear-sky conditions.
**3.3 Statistics of NPF events in the presence of ice particles**
The frequency of NPF occurrence in coincidence with ice particles is illustrated in Figure 4. The
upper panel (Figure 4a) exhibits the absolute occurrence frequency of number concentrations
$N_{uf}$ observed during NPF events. The graphs compile all measurements (more than 4600
samples of 1-Hz resolved data, cf. Table 1), which comply with the NPF criterion (black), for a
comparison with clear-air NPF events (green) and those, which were coincidently detected with
ice particles (red). At heights of in-cloud NPF observations (i.e. between 350 K and 380 K), the
number concentrations of particles larger than the ultrafine mode, i.e. $N_{15}$ and $N_{65}$, ranged (by
median) at $\sim$ 200 cm$^{-3}$ < $N_{15}$ < 1000 cm$^{-3}$ (COPAS) and $\sim$ 60 cm$^{-3}$ < $N_{65}$ < 150 cm$^{-3}$ (UHSAS-A,
Mahnke et al. (2020)). Two features are apparent:
1) Number concentrations $N_{uf}$ of more than $\sim$ 8000 cm$^{-3}$ seem more frequently observed
(about 1.5 times more often) in clear-air conditions. However, as the number of in-cloud
NPF observations with $N_{uf}$ > 8000 cm$^{-3}$ is comparably low ($\leq$ 10 encounters), the
statistics is likely insufficient for drawing additional conclusions from this. Whether or
not the presence of cloud ice confines the chance to detect very recent NPF (resulting in
high $N_{uf}$), is discussed in Section 6.

2) For NPF in the presence of cloud ice, number concentrations $N_{uf}$ between 1500 -
4000 cm$^{-3}$ were observed about twice as often as under clear-air conditions (Figure 4b).

Plausibly, highest $N_{uf}$ values are particularly found in the absence of deposition surfaces, which
ice particles would provide. It seems, however, less understandable why NPF should generate a
particular range of $N_{uf}$ more frequently in the presence of cloud ice. Further discussion on this
issue is provided in Section 6.
Until this point, the presence or absence of ice particles was distinguished by the criteria
$N_{ice}$ = 0 cm$^{-3}$ or $N_{ice}$ > 0 cm$^{-3}$, respectively. Figure 4b exhibits the occurrence frequency of $N_{uf}$ with
ice particles $N_{ice}$ > 0 cm$^{-3}$ normalised to the occurrence frequency of $N_{uf}$ of all NPF events (black
curve in Figure 4a). More than 75 % of observed NPF cases with 2000 cm$^{-3}$ < $N_{uf}$ < 4000 cm$^{-3}$
($\sim$ 200 samples) occurred while ice particles were present. In Figure 4c, the occurrence



frequencies of $N_{uf}$ are compiled for various levels of $N_{ice}$, which were normalised to $N_{uf}$ at
$N_{ice} > 0$ cm$^{-3}$ (red curve in Figure 4a). Thresholds of $N_{ice}$ are set with stepwise increasing number
concentrations (by one order of magnitude), to investigate whether the occurrence of NPF is
eventually confined or significantly influenced by the ice particle number density.
Although very faint, so called sub-visible cirrus clouds were found to comprise very small ice
particle number concentrations of $10^{-5}$ cm$^{-3}$ (corresponding to 0.1 per litre, cf. Kübbeler et al.
(2011); Spreitzer et al. (2017)). Sub-visible cirrus clouds with $N_{ice} < 10^{-3}$ cm$^{-3}$ are assumed to
have negligible influence on the NPF process, as is also to conclude from Figure 4c. Therefore, a
first threshold level is set to $N_{ice} > 10^{-3}$ cm$^{-3}$ (magenta curve), followed by a second threshold at
$N_{ice} > 10^{-2}$ cm$^{-3}$ (corresponding to 1 – 10 ice particles per litre, blue curve), which still represents
a comparatively small amount of $N_{ice}$ within sub-visible cirrus clouds (cf. Thomas et al. (2002);
Peter et al. (2003); Davis et al. (2010); Frey et al. (2011)). The maximum observed $N_{ice}$ reached
up to ~ 3 cm$^{-3}$. Concerning the frequency of observed $N_{uf}$, the difference between $N_{ice} > 0$ cm$^{-3}$
and $N_{ice} > 10^{-3}$ cm$^{-3}$ appears negligibly small. This leaves to conclude, that elevated $N_{uf}$ were
mostly observed coincidently with ice crystal number densities greater than $10^{-3}$ cm$^{-3}$. With
rising $N_{ice}$ level (above $10^{-2}$ cm$^{-3}$), the occurrence frequency of the highest $N_{uf}$ (> ~ 5000 cm$^{-3}$)
decreased. When $N_{ice}$ exceeds $10^{-1}$ cm$^{-3}$, the occurrence of $N_{uf} > 4500$ cm$^{-3}$ is significantly reduced
and $N_{uf} > 8500$cm$^{-3}$ were absent. At the highest observed $N_{ice}$ of ~ 3 cm$^{-3}$, NPF with $N_{uf} > 250$ cm$^{-3}$
were not detected anymore.
Hence, events with highest NPF-rate seem to occur preferentially at lower ice particle
concentrations or in clear air. At a certain $N_{ice}$ level (~ 3 cm$^{-3}$), the process of NPF seems to be
suppressed. This is in general agreement with earlier findings (Weigel et al., 2011), which
indicated the confinement of NPF by number densities above 2 cm$^{-3}$ of cloud ice particles with
diameter larger than 2 μm. Among other incidents, a singularly observed event was discussed
(*ibid.*), during which NPF was very likely suppressed by the excessive presence of cloud ice
particles, which then, on leaving the cloud, re-emerged with amounts of ultrafine particles of
almost previously observed magnitude. Although an ultimate observational evidence is
currently lacking, however, these findings suggest that NPF is entirely prevented in cases when
$N_{ice}$ substantially exceeds 2 - 3 cm$^{-3}$.
**4   In-cloud NPF related to *IWC* and cloud particle number densities**
**4.1 The relationship between cloud ice and aerosols**
Based on *in-situ* measurements over northern Australia and over West Africa, de Reus et al.
(2009) investigated the relationship between the number density of submicron aerosol particles
and the abundance of cloud particles at UT/LS levels. The authors provided aerosol and ice





particle number concentrations, which were averaged over the duration of various cloud
encounters in order to measure the fraction of submicrometre-sized particles that potentially
convert into cloud ice. Concerning the homogeneous ice nucleation process, a specific
relationship between the number concentration of aerosol and of ice particles cannot be
expected (Kärcher and Lohmann, 2002), whereas such a relationship is inherent in the ice
clouds' heterogeneous freezing process. From their analyses, de Reus et al. (2009) concluded
that a very similar range of ice-aerosol-ratios is observable in the convective outflow of ordinary
tropical convection (Australia) as well as of large, mesoscale convective systems (MCSs, West
Africa).
The measurements from StratoClim 2017 were compiled correspondingly to de Reus et al.
(2009) and are depicted in Figure 5. To ease the recognition of the relationship between the
measured number concentrations of ice particles and total aerosol ($N_{10}$), reference lines are
included in Figure 5, which indicate the number of encountered cloud particles per number of
submicrometre-sized aerosol particles. In addition to the density ratios of 1 : 300 and 1 : 30 000
(as in de Reus et al. (2009)), here also the 1 : 500 000 and the 1 : 5 000 000 ratios are marked.
The two panels in Figure 5 comprise the same data set of ice cloud encounters from
StratoClim 2017. The data were averaged over at least 10 seconds and over up to ∼ 23 minutes.
Several occasions were identified by de Reus et al. (2009) when comparatively high ratios with
up to a few hundreds of aerosol particles remained non-activated per single ice particle. The
cloud ice – aerosol – ratios, which were found in the Asian monsoon's convective outflow region,
are in general agreement with previous observations (de Reus et al., 2009) most of which were
limited to the blue shaded area in Figure 5. In agreement with previous findings, total aerosol
numbers of significantly less than a few hundreds per single ice particle were not observed
during StratoClim 2017. Up to $N_{10}$ of 700 cm$^{-3}$ almost all StratoClim data result from
measurements at mean ambient temperatures colder than -75 °C (correspondingly to the
observations by de Reus et al. (2009), shaded area). Frequent observations were made at
aerosol concentrations below 1000 cm$^{-3}$. Compared to previous findings, however, the
StratoClim data set comprises a lot more observations at cloud ice – aerosol - ratios between
1 : 3 000 and 1 : 500 000, including frequent events of elevated aerosol number concentrations
(> $10^3$ cm$^{-3}$). High total aerosol number concentrations of more than 6000 cm$^{-3}$, were observed
at $IWC$ values mostly below 10 µmol mol$^{-1}$ (i.e. log ($IWC$, µmol mol$^{-1}$) ≈ 1, Figure 5a). The majority
of observations were made at mean $IWC$ values below ∼ 300 µmol mol$^{-1}$ (i.e. log ($IWC$, µmol mol$^{-1}$) ≈ 2.5), which largely minimises the probability that the measured $N_{10}$ were impacted by
shattering artefacts from ice particles (cf. Appendix A). The majority of NPF occurrences (mostly
at ambient air temperatures between – 50 °C and – 80 °C) coincide with cloud ice – aerosol –





ratios between 1 : 3 000 and 1 : 500 000 (cf. Figure 5b). In particular, the abundance of in-cloud
NPF concentrates between ratios of 1 : 30 000 and 1 : 500 000, which may not further surprise,
as the large aerosol number concentrations are indicative to result from NPF. Concentrations $N_{10}$
of more than 1000 cm$^{-3}$ were not detected at ratios greater than 1 : 3 000. For $N_{10}$ above 500 cm$^{-}$
$^3$ and for cloud ice – aerosol - ratios smaller than 1 : 30 000, i.e. where elevated total aerosol
concentrations mostly coincide with lower ice particles densities ($\sim 10^{-3} – 10^{-1}$ cm$^{-3}$), the
observations predominantly occurred during NPF. However, cloud ice – aerosol – ratios greater
than 1 : 3 000 were reached mostly in the absence of NPF.
As pointed out by de Reus et al. (2009), there are caveats inherent with this kind of analyses. The
strength or efficiency of the aerosol activation is not straightforward to deduce from provided
ratios of total aerosol and cloud particle numbers. Many interdependencies exist that may
impact the illustrated relationship, such as
1) the altering of the aerosol particles (coagulation, condensation) or of the cloud elements

(sedimentation) or

2) the mixing of air masses with different aerosol and/or variable vapour saturation

characteristics (entrainment).

The ice formation process (*liquid origin* or *in-situ*) and the convection dynamics may additionally
affect the relationship of cloud elements and interstitial aerosol. Assigning ultrafine particles of
thousands per cm$^3$ (or more) to result from NPF is comparatively straightforward. In contrast,
$N_{uf}$ of a few 10 - 100 cm$^{-3}$ are potentially filtered by the NPF criterion, and are probably not
identified as NPF event, if detected at total aerosol concentrations ($N_{10}$) of comparable numbers.
Apart from demonstrating the reproducibility of earlier findings (de Reus et al., 2009), the
StratoClim mission allowed for extending this data set by new observations at different
conditions, particularly by including NPF.
In essence, these findings confirm that the occurrence of NPF is constrained by the cloud ice
microphysical properties such as particle size and number (both implied in the *IWC*). Total
aerosol number concentrations $N_{10}$ of a few hundreds per cubic centimetre were measured even
at highest cloud particle number concentrations ($N_{ice} > 2 \cdot 10^{-1}$ cm$^{-3}$) whereas, under such
conditions, NPF encounters remain exceptional. The following approach aims to narrow down
the cloud particle microphysical properties that limits the occurrence of in-cloud NPF.
**4.2 NPF in the *IWC-T* parameter space**
Analyses in earlier cirrus-related studies concerning the clouds' ice water content (*IWC*) as a
function of ambient air temperature provide insight into the processes inherent with the cirrus
formation (Krämer et al., 2016). As introduced by Luebke et al. (2016), Krämer et al. (2016), and
Wernli et al. (2016), a distinction of cirrus clouds regarding their formation mechanism is





obtainable within the *IWC-T* parameter space. The cirrus forms *in-situ* at elevated altitudes and
instantaneously at sufficiently cold temperatures. The *liquid-origin* cirrus cloud forms on
convective uplift from initially liquid droplets at lower altitudes (and less cold temperatures).
More specifically Wernli et al. (2016) distinguishes:
• *liquid-origin* cirrus: initially well-sized liquid cloud droplets freeze at almost

thermodynamic equilibrium in the ambient temperature range $235\,\text{K} < T < 273\,\text{K}$ under

nearly saturated conditions with respect to water (relative humidity $RH_w$ of $\sim 100\,\%$)

but at high supersaturation with respect to ice ($RH_i \gg 100\,\%$), while at freezing level, the

water is capable to coexist in each of its phases (vapour, liquid, and ice).

• *in-situ* cirrus: under exclusion of pre-existing large liquid cloud droplets, ice crystals

nucleate heterogeneously (due to deposition freezing) or freeze homogeneously from

tiny supercooled aqueous solution droplets (Koop et al., 2000), which are designated as

"too small to be considered as cloud droplets" (Wernli et al., 2016).

The main goal of juxtaposing *IWC* and ambient air temperature is to investigate differences in
the characteristics of ice clouds, which may influence the cirrus clouds' radiative properties.
Additionally, those cirrus clouds' properties can be investigated, which arise from the dynamics
and conditions in which the cirrus ice particles have formed.
In Figure 6 the *IWCs* versus ambient air temperatures are displayed for all cloud encounters
throughout StratoClim 2017 as a function (colour code) of
a) the mixing ratio of ultrafine particles (i.e. $n_{6-15} = n_{uf}$; Figure 6a),
b) the total mixing ratio $n_6$ of particles with $d_p > 6\,\text{nm}$ (Figure 6b) and
c) the CO mixing ratio (Figure 6c), respectively.
The upper panel of Figure 6 includes two data sets: (1) all data from StratoClim 2017 in 1-Hz
resolution (grey data points) and (2) only the resulting $n_{uf}$ complying with the NPF criterion
(colour coded data points). Mainly at very low ambient air temperatures ($\sim 200\,\text{K}$ and colder)
and for comparatively high *IWC* values, the $n_{6-15}$ (grey) data were available but many failed the
NPF criterion. The absolute values of the mixing ratio $n_6$ of submicrometre-sized particles were
relatively high (Figure 6b). The detection of likewise excessive mixing ratios $n_{15}$ (without
illustration) resulted in $n_{6-15}$, which did not exceed the threshold given with the NPF criterion (cf.
Section 1.1). Nevertheless, most of the $n_{6-15}$ data points, which failed the NPF criterion (cf. the
grey points in Figure 6a), coincide with the mixing ratios $n_6$ reaching up to several thousands of
$\text{mg}^{-1}$. It is not deducible from COPAS measurements how the enriched particle densities ($n_6$ and
$n_{15}$) distribute over the diameter spectrum of the submicrometre-sized aerosols. It therefore
remains open whether this observation is due to an expired NPF event with subsequent





coagulation of particles from the ultrafine size range (Weigel et al., 2020a), or whether the
particle enrichment is due to larger particles that were entered with the overshooting.
• The absence of NPF at excessively high *IWC* within very cold air (Figure 6) suggests that NPF is
confined as soon as strong overshooting prevails, due to the presence of predominantly *liquid-*
*origin* ice particles. Excessive *IWC* (> 1000 µmol mol$^{-1}$) at air temperatures colder than 200 K
indicates that strong, vertically overshooting convection had occurred. These high *IWC* most
likely originated from cloud ice, which had formed at lower levels from liquid droplets. The
amount of water vapour that is required to form ice clouds of comparable *IWC* at these air
temperatures is too large to explain the formation of encountered cirrus by another than the
*liquid-origin* process. This feature was observed during the StratoClim flights on 27 July and on
10 August 2017. Within the same temperature range ($T$ < 200 K), only a few NPF events with
moderately elevated $n_{uf}$ of more than ∼ 4000 mg$^{-1}$ (log ($n_{uf}$, mg$^{-1}$) ≳ 3.6, yellow and reddish
colours in Figure 6a) were encountered offside from strong vertical overshooting.
• In the presence of *in-situ* formed cirrus particles at cold temperatures, i.e. in or around the cold
point tropopause region, NPF events of remarkable strength ($n_{uf}$ > 5000, i.e. log ($n_{uf}$, mg$^{-1}$) > 3.7,
orange and reddish colours in Figure 6a) or very recent NPF bursts were rarely observed. When
the cloud ice has likely formed *in-situ* (CO < 80 nmol mol$^{-1}$, yellow, greenish and blue colours in
Figure 6c), NPF of reduced strength was observed ($n_{uf}$ < 1500 mg$^{-1}$, i.e. log ($n_{uf}$, mg$^{-1}$) < 3.2, bluish
colours of data points in Figure 6a). This indicates that NPF occurs in air with low CO content, i.e.
with comparatively low pollutant load.
• Suppression of NPF by cloud particles (due to the number and size of ice particles) could
explain why the number of ultrafine particles remained below the NPF criterion threshold at
comparatively high *IWC*, albeit the total mixing ratios ($n_6$ or $n_{15}$) were significantly elevated. It is
not likely that a high number of interstitial, non-activated aerosol is accountable for the
abundance of submicrometre-sized particles. The large particle quantities observed ($10^3$ -
$10^4$ mg$^{-1}$) and the comparatively moderate CO content of the air sampled (≲ 100 nmol mol$^{-1}$)
indicate a source of these particles at high altitudes. Very few hours after a completed NPF event
(≳ 4 h), however, the event may not be detectable anymore due to the short persistence of the
particles in the ultrafine size range (Weigel et al., 2020a). If the IWC values remained high over
several hours due to strong overshooting, and if NPF had happened more than four hours prior
to the measurements, then ultrafine particles could have coagulated to diameter sizes beyond
15 nm, hence, NPF would not have been identifiable anymore with COPAS.
• The air's low pollutant load is indicated by comparatively moderate or low CO mixing ratios
between 50 and about 100 nmol mol$^{-1}$ at ambient air temperatures of < 200 K (Figure 6c). For
comparison, the NPF observed during the West African monsoon were associated with CO levels
between 60 and 90 nmol mol$^{-1}$ (Weigel et al., 2011). Observation of moderate NPF
($n_{uf}$ < 1500 mg$^{-1}$, $\log(n_{uf}$, mg$^{-1}) \lesssim 3.3$) in the midst of *in-situ* formed cloud ice in air with
comparatively low pollutant load (CO < 80 nmol mol$^{-1}$) indicates that recent convective uplift of
polluted air is not a prerequisite for NPF to occur. Slow processes, which cause an accumulation
of NPF precursors at UT/LS altitudes, such as advection from elsewhere or the chemical and/or
photo-chemical conversion, likely suffice to supply a reservoir of precursor material. In air with
the highest CO content (> 100 nmol mol$^{-1}$), the *IWC-T*-values (for *T* > 200 K, i.e. at lower
altitudes) remain in expected ranges and they scatter within the limits of most frequent
observations (dashed black lines in Figure 6) as obtained from earlier analyses (Krämer et al.,
2016). At the highest CO content (> 100 nmol mol$^{-1}$), the $n_{uf}$ values remained predominantly
below 5000 mg$^{-1}$.

## 672    5    The dependency of NPF on the proximity to ice particles

### 673    5.1 NPF as a function of mean free distance between ice elements

Surfaces, such as those of ice particles, represent a potential sink for the gaseous precursor
species such as the $H_2SO_4$-$H_2O$ system, since the ice particles' coating (Bogdan et al. (2006);
Bogdan et al. (2013)) offers the necessary attachment points for the molecules of a condensable
vapor. Consequently, an abundance of condensation surface should reduce or even prevent the
NPF process. Cloud ice particles provide a comparatively large surface for coating, which raises
the question whether NPF is affected by the presence of these particles.
The free distance between the ice particles is quantified based on the measurements of $N_{ice}$ and
of the ice particles' mean mass radius $\overline{r_{ice}}$, (consider $\overline{r_{ice}}^3 \sim \frac{IWC}{N_{ice}}$). The mean free volume in between
the ice particles (the inter-crystal volume, *ICV* per cm$^3$ of air) is calculated with the number $N_{ice}^*$
of ice particles per air volume (instead of the particles' number concentration) as:
$$ICV = \frac{V - \frac{4}{3}\pi \cdot \bar{r}_{ice}^3 \cdot N_{ice}^*}{N_{ice}^*} \qquad (2),$$

which basically subtracts the total ice volume from the sampled air volume (*V* = 1 cm$^3$) and the
division by $N_{ice}^*$ yields the *ICV*. Consequently, the *ICV* represents the mean particle-free volume
assuming the distribution of ice crystals within the air volume as homogeneous. As long as the
particle number and size remain small, subtracting the total ice volume from the air volume in
equation (2) yields results without significant contribution. With a maximum of measured
$N_{ice}^*$ = 3 cm$^{-3}$ together with the maximum detected ice particle radius of 100 μm, the subtraction
$V - \frac{4}{3} \cdot \pi \cdot \bar{r}_{ice}^3 \cdot N_{ice}^*$ corresponds to 1 cm$^{-3}$ – 10$^{-11}$ cm$^{-3}$. Hence, the volume of ice is insignificant



compared to the volume of air, and the *ICV* may be considered as a function of $N_{ice}$ only. The
mean inter-crystalline distance (*ICD,* in cm) is then calculated by:
$$\boldsymbol{ICD} = \sqrt[3]{\frac{\boldsymbol{ICV}}{\left(\frac{4}{3}\cdot\pi\right)}} \qquad (3),$$

and the *ICV* is assumed as a sphere around every individual ice particle. The radius of each
sphere constitutes the mean ice-free distance into any direction from the individual ice particle.
Conceptually, this approach corresponds to the definition of the cloud elements' distance
provided by Baumgartner and Spichtinger (2018).
Figure 7a depicts the number concentration of ultrafine particles ($N_{uf}$) as a function of the
calculated ice particles' mean free distance from each other. The continuous colour transition of
the data points from red to blue indicates the independence of the number of ultrafine particles
in reference to the ice particles' mean free distance and rather documents the obvious
relationship between the number of ice particles and their distance. The present ice particles
compete for the limited amount of available water vapour; consequently, elevated number
concentrations of ice particles are associated with many small ice particles. In essence, only the
number of ice particles $N_{ice}$ would not be able to constrain the occurrence and/or strength of
NPF, as under encountered atmospheric conditions, a wide scattering of $N_{uf}$ concentrations was
observed at any *ICD* between about 1 cm and 10 cm.
Figure 7b shows the ice particles' mean mass radius $\overline{r_{ice}}$ as a function of the *ICD* and the number
of ultrafine particles. By means of the mean mass radius $\overline{r_{ice}}$, two different cases were
distinguished:
a) In the smallest ice particle size range ($\sim 3\,\mu m < \overline{r_{ice}} < \sim 20\,\mu m$, $\log(\overline{r_{ice}}, \mu m) \lesssim 1.3$), a

dependency of the *ICD* on the particle size was discernible. For instance, smallest ice

particles (bluish $\overline{r_{ice}}$) predominantly coincided with short *ICD* of about 1 cm at elevated

$N_{ice}$. Towards larger *ICD*, ice particle sizes continuously increased up to $\overline{r_{ice}} \approx 20\,\mu m$,

which again reflects the competition of the ice crystals for the available water vapour.

However, within the same interval of ice particle sizes ($\overline{r_{ice}} < \sim 20\,\mu m$), the

concentrations $N_{uf}$ scattered over almost two orders of magnitude (from $\sim 100\,cm^{-3}$ to

$\sim 10\,000\,cm^{-3}$) up to *ICD* of $\sim 10$ cm without any obvious systematic.

b) In the presence of larger ice particles, $\overline{r_{ice}} > \sim 30\,\mu m$ ($1.3 < \log(\overline{r_{ice}}, \mu m) \lesssim 1.4$, orange

and reddish colours), the *ICD* ranged from $\sim 1$ cm to values above $\sim 10$ cm. Hence, not

only $\overline{r_{ice}}$ determined the resulting *ICD,* but $N_{ice}$ increasingly contributed as well.

Unexpectedly, the concentrations $N_{uf}$ were not at the highest when *ICD* values reached

their maximum of slightly more than 10 cm. For largest particles sizes ($\overline{r_{ice}} > \sim 30\,\mu m$),





the values of $N_{uf}$ accumulate at number concentrations of $\sim$ 400 - 4000 cm$^{-3}$ over the
entire range of *ICDs*.
As long as the mean ice particle radius remained below a few dozen µm, NPF was encountered
with almost any resulting $N_{uf}$ concentration. It was shown before (Section 4.4 and Figure 4), that
a wide scatter of $N_{uf}$ was observed largely independent from coincidently detected number $N_{ice}$
of ice particles. Hence, in-cloud NPF – as found during StratoClim 2017 - occurred almost
unaffected by the ice particle number, as long as the mean ice particle size remained small
enough (i.e. with $\overline{r_{ice}}$ < 20 µm).
Instead of evaluating the number of ultrafine particles as an exclusive function either of ice
crystal number or of the ice particle radius, respectively, the *IWC* combines both microphysical
parameters of the observed ice clouds, particle size and number concentration. The particle
mass (i.e. the particle radius to the third power, $r^3$) is proportional to *IWC* and $N_{ice}$. Indeed, if $N_{uf}$
over *ICD* are analysed as a function of *IWC,* a certain systematic becomes visible (Figure 7c). At
lower *IWC* (< 1 µmol mol$^{-1}$, log (*IWC*, nmol mol$^{-1}$) $\lesssim$ 0, bluish and green colours) the *ICDs* were at
the largest and observed NPF was of the highest intensity ($N_{uf}$ of several thousands per cm$^3$).
Between 1 µmol mol$^{-1}$ and 10 µmol mol$^{-1}$ (yellow colours) the maximum of $N_{uf}$ throughout
observed NPF events was reduced. The maximum $N_{uf}$ was further reduced when *IWC* further
increased beyond 10 µmol mol$^{-1}$. This result shows that the maximum $N_{uf}$ reached throughout in-
cloud NPF was determined by the combination of both, the ice particles' number concentration
$N_{ice}$ and their mean mass radius $\overline{r_{ice}}$ .
**5.2 NPF as a function of cloud elements' integral radius *IR***
Indications were found that both, number density and size of cloud ice particles, have a
complementary effect on the amount of ultrafine particles ($N_{uf}$) resulting from in-cloud NPF. This
motivates the compilation of $N_{uf}$ values as a function of the integral radius $IR = \overline{r_{ice}} \cdot N_{ice}$ of the
ice particle population. The parameter *IR* was described, e.g., by Manton (1979), or Politovich
and Cooper (1988), and is frequently used to characterise clouds' microphysical properties (e.g.
Korolev and Mazin (2003); or Krämer et al. (2009)). *IWC* and *IR* are expected to be strongly
related as the diffusive growth of an ice particle is proportional to *IR* (see e.g. Pruppacher and
Klett (2012)). The relationship between *IWC* and *IR* is also apparent from a systematic sorting of
the data points displayed in Figure 8a. The probability should be high that weak NPF (generating
low $N_{uf}$) often occurred in the presence of ice particles. In contrast, the occurrence of excessive
NPF events in the cloud (with $N_{uf}$ significantly exceeding several thousand per cm$^3$) was less
likely. For almost all *IR* below 1 µm cm$^{-3}$, however, the $N_{uf}$ concentrations were unsystematically
scattered over the entire interval between $\sim$ 100 cm$^{-3}$ and $\sim$ 10 000 cm$^{-3}$.





Towards the highest *IR* (> 1 μm cm$^{-3}$), the maximum of observed $N_{uf}$ continuously decreased.
Generally, this may reflect a limiting influence by the cloud ice on the maximum strength of
occurring NPF (indicated by the diagonal grey-shaded bars in Figure 8). An exceptional feature
is exhibited in Figure 8a with a high signal of $N_{uf}$ (~ 3000 – 4000 cm$^{-3}$) amongst elevated *IR*
(between ~ 4 and 10 μm cm$^{-3}$). This cluster of data points resulted from the measurements of
two individual mission flights, on 27 July (~ 3000 cm$^{-3}$ < $N_{uf}$ < ~ 3500 cm$^{-3}$) and on 06 August
(~ 3500 cm$^{-3}$ < $N_{uf}$ < ~ 4000 cm$^{-3}$), respectively. During these measuring periods, ice particle
densities ($N_{ice}$) and the mean ice particle sizes (i.e. the particles' mean mass radius $\overline{r_{ice}}$ ) did not
rise above 0.1 - 0.3 cm$^{-3}$ and 25 - 50 μm. Neither $N_{uf}$ nor the ice microphysics exceeded the range
of moderate values. The two independent exceptional observations may indicate a
local/temporal state of imbalance and could have been caused by:

1)  moderate NPF, which was just proceeding when measured or which had been completed

very recently (in such a case, the observed $N_{uf}$ should rapidly decay due to coagulation,

within less than one hour, to values of ~ 1000 cm$^{-3}$), or

2)  ice particles, which sediment from high altitudes into an area of currently active NPF,

3)  cooling of air accompanied with nucleation of ice, while the cooling is due to air parcel's

vertical displacement possibly resulting from convective overshooting or gravity wave

activity (cf. Weigel et al. 2020a).

The generally limiting influence by the cloud ice on the maximum strength of NPF, that is
indicated by the majority of observations, is possibly explainable by the reduction of NPF
precursors due to condensation onto present surfaces provided by the ice particles (maximum
$N_{ice}$: 2 - 3 cm$^{-3}$). The question arises whether the distance between the ice particles allows
efficient absorption and sustained reduction of NPF precursor molecules, or whether such an
effect is only likely in the immediate vicinity of an ice particle. However, the effectiveness of such
a process strongly depends on the diffusivity of the NPF precursor's molecules. If the molecules
of the main NPF precursor are absorbed before thermodynamic conditions for NPF are reached,
then these molecules are removed and missing in the formation of molecular clusters as initial
step in the nucleation process. Sulphuric acid ($H_2SO_4$) is one of the most prominent condensable
vapours and NPF precursors in the atmosphere. Numerical analyses concerning the reduction of
the saturation ratio of $H_2SO_4$ due to the presence of ice particles, which are coated with $H_2SO_4$ (as
typical for cirrus particles at 10-20 km altitude; cf. Bogdan et al. (2006); Bogdan et al. (2013))
are described in Appendix B: Impact of ice particles on NPF precursors' saturation ratio(see also
Figure B- 1). Although the binary $H_2SO_4$-$H_2O$ nucleation process alone is assumed as insufficient
to explain atmospheric NPF (Bianchi et al. (2016); Kirkby et al. (2011)), the numerical analysis
may qualitatively apply also for saturated condensable vapours containing compounds other
than dissolved $H_2SO_4$ (Riccobono et al., 2014).





The numerical analysis yielded that the precursor's saturation ratio decreases rapidly with
increasing $IR$. As long as the ice particles' size remains small (radii < 10 µm) their influence on
the saturation ratio of the NPF precursor is comparatively weak. However, as demonstrated for
$H_2SO_4$ (cf. Appendix B), rising $IR$ (combining ice particle size and number) could crucially confine
the production of high $N_{uf}$, or inhibit NPF at all. In particular, only completely uncoated ice
particles of pure water (which are excluded to exist in the UT/LS; cf. Bogdan et al. (2006);
Bogdan et al. (2013)) would be ineffective condensation surfaces for $H_2SO_4$ vapour, since
attachment points for $H_2SO_4$ molecules were lacking on the surface of pure ice water. Hence, the
frequent observations of in-cloud NPF is indicative for processes, which are capable of
maintaining sufficiently high NPF precursor saturation ratios. Such processes could involve
turbulent mixing of precursor-enriched air (entrainment) or a cooling process as induced, e.g. by
a temperature anomaly due to gravity wave activity (cf. Weigel et al. (2020a)). Otherwise, NPF
observations should be less frequently observable in the view of ice particles' effective influence
on the saturation ratio of NPF precursors.
From the results shown in Figure 8a, it may be concluded that the $N_{uf}$-range of 500-3000 cm$^{-3}$ is
most frequently observed over the entire extent of detected $IR$ values. While this confirms the
impression from Figure 4 (cf. Section 4.4), the conclusions from Figure 8 allow for approaching a
possible explanation of the $N_{uf}$'s behaviour with $IR$:

1)  The maximum $N_{uf}$ resulting from in-cloud NPF is determined by $IR$. Abundant ice

particles of sufficient size are capable of reducing the saturation ratio of NPF precursors

within time scales ranging from half an hour to a few hours. Consequently, moderate or

weak NPF events with less excessive $N_{uf}$ production may occur most frequently in the

presence of cloud ice. However, the probability to instrumentally identify weak events

decreases with decreasing $N_{uf}$.

2)  Furthermore, coagulation also affects $N_{uf}$ in time scales of a few to dozens of hours (cf.

Weigel et al. 2020a), very likely constituting the most efficient altering process of

ultrafine particles from NPF.

At the time of observation, the age and processing progress of the ultrafine particles are
unknown. Amongst the previously described effects, the temporal delay between the NPF event
and the measurement may have a crucial but unquantifiable impact on the actually observed $N_{uf}$,
as the altering of ultrafine particles is very effective in time scales of a few hours (Weigel et al.
2020a). Hence, it is likely a matter of probability, that in-cloud NPF with moderately high $N_{uf}$
(e.g. 500-3000 cm$^{-3}$) is most frequently observed. According to the data compiled in Figure 8, $IR$
values of about 24 µm cm$^{-3}$ (corresponding to $N_{ice}$ of about 0.7-0.8 cm$^{-3}$ and mean mass radii $\overline{r_{ice}}$
of about 32 µm) constituted in general the uppermost limit for in-cloud NPF observation during
StratoClim 2017.
Another processing of the same data set, i.e. $N_{uf}$ as a function of $IR$, implies a data sorting by
means of CO mixing ratio (Figure 8b). Apparently, none of the emerging samples, neither with
highest $N_{uf}$ nor with highest $IR$, was directly ascribable to polluted air, which was recently lifted
from the surface. Strongest NPF ($N_{uf} > 5000$ cm$^{-3}$) were exclusively observed at CO mixing ratios
ranging between $\sim 90$ and $100$ nmol mol$^{-1}$, which indicates the air's moderate pollutant load or
its moderate age. Alternatively, these CO values may reflect certain mixing states of air masses of
significantly different age. In less polluted air (CO mixing ratios below$\sim 70$ nmol mol$^{-1}$), the $IR$
reaches the highest values (up to $\sim 24$ µm cm$^{-3}$) which were observed together with elevated
$IWC$ (up to $\sim 750$ µmol mol$^{-1}$, i.e. log ($IWC$, nmol mol$^{-1}$) $\approx 0.88$). Within pristine air, cloud ice
particles mostly likely form *in-situ*. It is conceivable, that the *in-situ* cloud ice formation and NPF
happens simultaneously, potentially induced by the same process: e.g. by updraughts due to
subjacent convection (pileus effect) or by (local) cooling due to gravity waves (cf. Weigel et al.
2020a). In such cases (CO $< 70$ nmol mol$^{-1}$), the observed $N_{uf}$ are systematically lower than
$1000$ cm$^{-3}$ and they mostly range at a few hundreds per cm$^{3}$.
Hence, air masses with low pollutant loads obviously contain sufficient amounts of precursor
material to supply moderate NPF ($100$ cm$^{-3} < N_{uf} < 1000$ cm$^{-3}$) which may strengthen the
hypothesis that air's pollutant load is not an essential prerequisite for the occurrence of most
intense NPF ($N_{uf} > 5000$ cm$^{-3}$ at $IR < 1$ µm cm$^{-3}$) in the UT/LS region. This differs from earlier
findings from ground-based measurements at high mountain sites (at about 5 km altitude) in the
Himalaya region (Venzac et al., 2008) or at the Jungfraujoch station ($\sim 3.5$ km altitude) in the
Swiss Alps (Bianchi et al., 2016) who attributed their frequent NPF observations to the
advection of polluted air which rises up from the valleys towards the research stations.
Williamson et al. (2019) made their very frequent NPF observations based on a very
comprehensive data set of airborne *in-situ* measurements over both oceans, the Atlantic and the
Pacific, i.e., in certain distance away from direct convective supply by industrial pollution.
However, different atmospheric conditions and/or different chemical precursor species might
play a role in the NPF processes occurring in the boundary layer or at UT/LS altitudes.
## 6 Summary and Conclusions
Between 27 July and 10 August 2017 the airborne StratoClim mission took place in Kathmandu,
Nepal, comprising eight mission flights ($\sim 22.5$ hours of COPAS measurement time above 10 km,
$\theta \gtrsim 350$ K) up to altitudes of 20 km ($\theta \approx 475$ K) with the Russian high-altitude research aircraft
M-55 *Geophysica*. The present analysis includes the description and discussion of New Particle
Formation (NPF) in the presence of cloud ice particles as observed in the UT/LS region of the
Asian Monsoon Anticyclone (AMA) over northern India, Nepal and Bangladesh. Elevated
concentrations of ultrafine particles ($N_{uf}$) generated by NPF were observed in hitherto
unexpected frequency together with ice particles ($N_{ice} > 0$ cm$^{-3}$) at altitudes between $\sim 11$ km


and 16 km (~ 355 - 385 K) and mainly at ambient temperatures colder than ~ 230 K. During
StratoClim 2017, a total number of 104 in-cloud NPF events was observed over a total duration
of 1 hour and 17 minutes (~ 5 % of the total data set, ~ 49 % of all observed NPF cases).
Maximum concentrations of ultrafine particles of up to ~ 11000 cm$^{-3}$ ($\approx$ 50000 mg$^{-1}$) were
detected coincidently with ice particles in concentrations $N_{ice}$ of 0.05 – 0.1 cm$^{-3}$ (correspondent
to 50 - 100 ice particles per litre) at heights of approximately 15 km (~ 370 K).
Analyses of the StratoClim data set concerning the relationship between interstitial aerosol and
the abundance of cloud particles in the UT/LS are consistent with the findings from earlier
measurements (de Reus et al., 2009), and extended these by new observations under different
conditions. When ice particles are abundant ($N_{ice}$ > 0.5 cm$^{-3}$), total aerosol number
concentrations ($N_{10}$) remain generally between ~ 200 cm$^{-3}$ and 700 cm$^{-3}$. In agreement with
earlier findings (de Reus et al., 2009), the ratio of ice particle number and the number of
submicrometre-sized aerosols did not significantly rise above 300 aerosols per ice particle at
low air temperatures (< 200 K). Intense NPF, generating ultrafine particles of several thousands
per cm$^{3}$, decrease the ice particle – aerosol – ratio substantially. However, such intense NPF was
not observed at ratios larger than 1:3000, which indicates that the presence of cloud ice imposes
limitations to the occurrence of NPF.
In-cloud NPF appears confined in the presence of predominantly *liquid-origin* ice particles with
increased ice water content resulting from strong convective overshooting. This is confirmed by
coincidently measured CO content of the air sample: air's pollutant load and/or its recent
surface contact do not determine the strength of in-cloud NPF. Otherwise, the most intensive
NPF events should occur within air masses with highest CO content. When the cloud ice has
formed *in-situ,* at low CO mixing ratios, NPF was observed although with reduced strength.
However, it is not yet conclusively clarified whether the direct convective supply of precursor
material from pollution in the boundary layer is an essential prerequisite for the occurrence of
NPF in the UT/LS, or whether NPF together with the ice cloud formation are initialised in
processed and diluted air masses. The observations suggest that sufficient amounts of NPF
precursor accumulate at UT/LS altitude, which is not necessarily connected to air's recent
vertical uplift. It remains speculative, and it should be subject of suitable numerical analyses, to
which extent the vertically lifted ice particles themselves contribute as carrier for soluble NPF
precursor gases such as $SO_2$, $H_2SO_4$, or others, e.g., if dissolved in the cloud elements' liquid
phase at lower heights and if released again at TTL altitudes after the cloud ice has sublimated.
Comparatively slow processes, as air mass transport from elsewhere or the chemical and/or
photo-chemical conversion at elevated altitudes may suffice to supply the reservoir of NPF
precursors at UT/LS altitudes. NPF of highest intensity, however, was observed at moderate CO


mixing ratios, indicating a moderate pollutant load, and a certain age or mixing state of the air
mass. Intense NPF seems almost confined in strong convective updraughts (cf. Section 4.2),
either because of the intense dynamics inherent with overshooting convection, or because the
precursor's saturation ratio of recently uplifted air does not suffice for immediate NPF.
The occurrence of NPF is strongly dependent on the precursor's saturation ratio. Ice particles in
sufficient number and size are well capable to reduce the saturation ratio of a NPF precursor
such as $H_2SO_4$. This implies two conclusions: 1) in-cloud NPF is confined by abundant ice
particles and 2) not only the number of ice particles limits the NPF occurrence but also the ice
particles' size. The strength of in-cloud NPF most clearly depends on the integral radius $IR$ (=
$\overline{r_{ice}} \cdot N_{ice}$), the product of the ice particles number concentration and the ice particles' mean
mass radius. The $IR$ turned out as appropriate cloud ice related parameter to juxtapose with NPF
data. Up to $IR$ of $\sim 1\,\mu m\,cm^{-3}$ the occurrence of NPF of any strength (with
$\sim 100 < N_{uf} < 10\,000\,cm^{-3}$) seems independent on the presence of ice particles at all. At larger $IR$
($> 1\,\mu m\,cm^{-3}$) the presence of ice particles limits the maximum of $N_{uf}$ from NPF. This result
refines earlier findings (Weigel et al., 2011) that mainly the number of ice particles would limit
the occurrence of NPF.
The observations indicate that a $N_{uf}$-range of 500-3000 $cm^{-3}$ was most frequently observed
during in-cloud NPF. However, weak NPF generating only low $N_{uf}$ may occur most frequently in
the presence of cloud ice, whilst the probability to instrumentally identify such weak events
decreases with decreasing $N_{uf}$. Additionally, coagulation affects large $N_{uf}$ in time scales of a few to
dozens of hours (cf. Weigel et al. 2020a). As a consequence, the supposedly preferred $N_{uf}$-range
likely results from superimposed effects, and it may be a matter of probability and timing (delay
between NPF event and observation) that the $N_{uf}$-range of 500-3000 $cm^{-3}$ is most frequently
observed in the presence of cloud ice.
At the moment of observation, the age of the ultrafine aerosol (the delay between the NPF burst
and the instrumental detection) as well as the aerosol's processing history is unknown. While
the aerosol's persistence in the ultrafine size range is limited, it is conceivable that the
abundance of aerosols influences the local formation of ice particles, or that ice particles are
coated by ultrafine aerosol material due to coagulation. Above certain sizes, the cloud ice
elements are increasingly subject to sedimentation. At warmer ambient temperatures, the ice
particles may sublimate. This could release the materials attributed to the initially NPF-
generated ultrafine aerosol. It remains speculative whether or not, in terms of physico-chemical
characteristics, the released aerosol material is comparable with the primary NPF-generated
aerosol. However, the sublimation of coated ice particles and the release of aerosol material at
intermediate altitudes could provide nuclei for entrainment and/or cloud formation. It remains





unquantified, however, whether NPF near the surface (cf. Venzac et al. (2008) or Bianchi et al.
(2016)) or the NPF at UT/LS altitudes contribute at the most to the availability of cloud
condensation nuclei (CCN), which are supposed to promote cloud formation (Andreae et al.,
2018) at the cloud condensation levels. Most likely the specific source contributions to the
abundance of available CCN are as variable as are the chemical species potentially involved in
the NPF process.
Data availability:
*The data shown in this study are available at the StratoClim campaign database at*
https://stratoclim.icg.kfa-juelich.de/AfcMain/CampaignDataBase
*or they may be provided by respective PI upon request.*
Author contribution
*RW evaluated the data, created the figures, and draughted the manuscript with contributions by CM, MB, MK,*
*HT and PS. SB participated in the data analyses and the manuscript draughting. Numerical simulations*
*concerning the impact of ice particles on the saturation ratio of $H_2SO_4$ were performed by MB with contributions*
*by HT. MK, NS, AA and CR contributed with cloud microphysical and water vapour data, SV and FD'A took*
*care of the CO data. The manuscript was critically reviewed by CM, MB, MK, PS, NS, AA, CR, SV, FD'A, HT,*
*and SB.*
Competing interests
*The authors declare no competing interests.*
**Acknowledgements**
The contributions from the workshops of the Max Planck Institute for Chemistry and of the
Institute for Physics of the Atmosphere (Mainz University) were essential for this work. In
particular, we acknowledge support of T. Böttger, M. Flanz, C. v. Glahn, H. Rott, and W. Schneider.
Also acknowledged are the comprehensive and helpful discussions with M. Szakáll. We very
much thank the crew of MDB (Myasishchev Design Bureau) and the M-55 *Geophysica* pilots. The
extraordinary commitment of F. Stroh in realisation of the campaign and the leadership of the
entire StratoClim project by M. Rex are gratefully acknowledged. Some of our research leading to
the presented results received funding from the European Research Council under the European
Union's Seventh Framework Program (FP/2007-2013)/ERC Grant Agreement No. 321040
(EXCATRO). The StratoClim project was funded by the EU (FP7/2007–2018 Grant No. 603557)
and also supported by the German "Bundesministerium für Bildung und Forschung" (BMBF)
under the joint ROMIC-project SPITFIRE (01LG1205A). M. Baumgartner acknowledges support
by the DFG within the Transregional Collaborative Research CentreTRR165 "Waves to Weather",
Project Z2. P. Spichtinger acknowledges support by the DFG within the research unit Multiscale
Dynamics of Gravity Waves (MS-GWaves) through grant SP 1163/5-2. H. Tost acknowledges
funding from the Carl-Zeiss foundation. We explicitly thank the officials of the Nepalese
government authorities, research institutions and Tribhuvan Airport as well as of the German





Embassy for their extraordinary support and hospitality, which enabled our field campaign and
research.

**Appendix A: Exclusion of artefacts on NPF observation due to the presence of cloud ice**

During the herein discussed NPF events, the detected total number concentration of cloud
elements never exceeded ∼2-3 cm$^{-3}$. Thus, the number density of cloud elements were always
significantly smaller (at least by two orders of magnitude) compared to detected aerosol
number concentrations. At ambient air temperatures much colder than 235 K (and as cold as
187 K), the clouds entirely consisted of ice particles. In other studies, however, the discussions
on NPF are restricted to measurements under cloud-free (clear-air) conditions as the cloud
particles are suspected to possibly impact onto the aircraft's hull or the aerosol inlet, this way
possibly generating artefacts on the aerosol measurements (Williamson et al. (2019) referring to
Weber et al. (1998)). Regarding the in-cloud NPF observations throughout StratoClim 2017, the
following aspects are noteworthy:
1) At typical flight speeds of the M-55 *Geophysica* (154 ± 39 m s$^{-1}$), sub-micrometre-sized
ice particles should negligibly be subject to impaction on parts of the aircraft structure
(nose, wing's leading edge, etc.) as the particles are well capable to follow the air stream
around such flow obstacles (Kulkarni et al., 2011). Furthermore, ice particles in the
diameter size range of a few micrometre (i.e. 1 µm < $d_p$ < 10 µm) likely sublimate in the
congestion region upstream of any aircraft structure (e.g. the wings leading edge, or the
aerosol inlet). Even though a single particle of the aforementioned size could randomly
enter the COPAS aerosol inlet, the diffuser-type entry of the aerosol inlet leads to a
deceleration of the air flow inside the probe head (Weigel et al., 2009) accompanied with
a sudden temperature increase in the air sample (according to fluid dynamical
simulations of the inlet flow; Weigel et al. (2009) and references therein). Hence, rapid
sublimation of ice particles in the diameter size range of a few µm can be expected to
occur inside the aerosol inlet of COPAS. The entry of the sample air into the inlet's
second diffuser additionally reduces the sampling of ice particle fragments.
2) The number concentration of ice particles with diameter $d_p$ > 10 µm mostly remained
below 0.4 cm$^{-3}$ when coincidently detected with NPF. On impact and shattering of a
single ice particle of such a size, the number of generated fragments is estimated to
range at about 10-100 per cm$^3$ (Korolev et al., 2013). Hence, to substantially affect the
detected number concentration of ultrafine particles (on magnitude order of hundreds to
up to ten thousands per cm$^3$), the number of ice particles possibly emanating from
shattering appears too low.
3) The probability that ice particles hit the sharp edged tips of the COPAS aerosol inlet
(Weigel et al., 2009) appears negligibly small. The impaction surface provided by the





COPAS aerosol inlet is mainly the inlet's ring-shaped entry with an opening diameter of
$\sim 7.3$ mm and a wall thickness of $\sim 100$ μm. In the unlikely case that a single ice particle
impact occurred, all generated fragments were required to endure the temperature rise
within the inlet head (cf. first argument of this list) and the transport through the aerosol
lines towards the COPAS detectors before they can cause any effect on the measurement.
An effect of shattered large ice particles on the detection of ultrafine aerosol particles is
ultimately not excludable. However, despite the reference by Williamson et al. (2019) in this
context, ice particle fragmentation was not described by Weber et al. (1998). The same authors
discuss the influence on NPF detections due to fragmentation of supercooled liquid-water cloud
droplets and suggest a careful discussion in such cases. In general, such an influence due to the
fragmentation of ice particles was largely ruled out or estimated as much lower than that of
liquid droplets (Weber et al., 1998). Concerning the analyses discussed herein, however, it
seems a statistical exception that ice particle fragments emanating from shattered ice particles
crucially affect the measurement of the numbers of ultrafine aerosol particles. Moreover, if the
NPF detections were systematically affected by the presence of cloud ice, the observed
quantities of ultrafine particles would probably feature systematic and larger differences during
in-cloud measurements compared to clear-air observations. None of the described artefacts was
observable in the data from StratoClim 2017.
**Appendix B: Impact of ice particles on NPF precursors' saturation ratio**
Calculations were made regarding the time scales in which the decrease of the supersaturation
of $H_2SO_4$ vapour occurs in the presence of coated ice particles. These serve as estimates
concerning the efficiency of the diffusional loss of condensable materials, i.e. of the process
competing with the gas-to-particle conversion of these vapours. The molecules of condensable
and saturated (or supersaturated) vapours condense onto available surfaces, such as provided
e.g. by an ice particle, whereas the combination of molecules into stable clusters requires
significantly supersaturated conditions to form new particles out of the gas phase. However, it
seems plausible that in the closest vicinity of an ice particle the condensational loss of a
precursor gas like sulphuric acid ($H_2SO_4$) predominates over the NPF process. For $H_2SO_4$, as a
representative of the NPF precursors, the question arises how efficient the condensation of
$H_2SO_4$ occurs onto provided surface. The molecules' mobility and the condensation efficiency of
the $H_2SO_4$ molecules is mainly determined by their diffusivity under the given atmospheric
conditions. The diffusivity of $H_2SO_4$ is about a factor of 0.2-0.5 of the diffusivity of water vapour
(Tang et al., 2014). Consequently, the condensational deposition of $H_2SO_4$ onto the coated
particles surface causes the saturation ratio of $H_2SO_4$ to decrease within the environment of the



ice particle, which likely suppresses the process of NPF within a certain range around the ice
particle.
Presuming that the ice particles are coated with $H_2SO_4$ (Bogdan et al. (2006); Bogdan et al.
(2013)), model simulations were performed to investigate the timescales within which the
coated ice particles reduce various $H_2SO_4$ saturation ratios. The simulation results (shown in
Figure B- 1) are based on constant ambient temperature ($T \approx 200$ K) and pressure ($p = 110$ hPa)
conditions. For the same temperature conditions, the saturation vapour pressure $p_{sat}$ of $H_2SO_4$ is
calculated according to Vehkamäki et al. (2002). This way, the degree of supersaturation is
deducible from the $H_2SO_4$ molecules concentrations reported for the CLOUD (Cosmics Leaving
OUtdoor Droplets) chamber experiments (cf. Kürten (2019), and references therein). According
to this study, and in agreement with other references (H. Gordon, School of Earth and
Environment, Leeds University, UK, personal communications Oct. 2019), molecule
concentrations of $10^6 - 10^7$ cm$^{-3}$ are required in the CLOUD chamber at temperatures of 208 K
to induce NPF with nucleation rates of $10^{-2} - 100$ cm$^{-3}$ s$^{-1}$ (read out from Fig. 4 in Kürten
(2019)). Keeping possible wall effects of the laboratory experiments in mind, for the occurrence
of NPF under real atmospheric conditions, the lower bound of required molecule concentrations
($10^6$ cm$^{-3}$) may suffice, with an uncertainty of a factor five (H. Gordon, School of Earth and
Environment, Leeds University, UK, personal communications Oct. 2019). At an ambient
temperature of 208 K, the molecule concentrations of $10^6 - 10^7$ $H_2SO_4$ cm$^{-3}$ (Kürten, 2019)
correspond to saturation ratios of about $S \approx 10 - 100$. The following analysis, however,
comprises a much wider range of saturation ratios between 10 and up to 5000 to account for a
higher sensitivity of the temperature dependency of $S$ and for other nucleation rates than chosen
for this study.
Based on the expression formulated by Tsagkogeorgas et al. (2017) with the saturation vapour
pressure $p_{sat}$ of $H_2SO_4$ (above a flat surface) and with an accommodation coefficient of $\alpha = 0.65$
(Pöschl et al., 1998), the ice particle's change of mass $m$ per time unit is calculated by:
$$\frac{dm}{dt} = \frac{4\pi Dr(S-1)}{\left(\frac{L}{RT}-1\right)\frac{L}{T}\frac{D}{K}+\frac{RT}{\alpha \cdot p_{sat}}} \qquad (B-1),$$

which conceptually represents the change of mass (size) of the particles, onto which the $H_2SO_4$
condenses and which is also consistent with the finding that cirrus cloud elements are coated
with a $H_2SO_4$-$H_2O$ layer (Bogdan et al. (2006); Bogdan et al. (2013)). The diffusivity of $H_2SO_4$
molecules in air is denoted with $D$, and $K$ refers to the thermal conductivity of air, while $R$ and $R_a$
are the gas constants of $H_2SO_4$ and the air, respectively. However, since the particle growth of
micrometre-sized ice particles due to condensation of $H_2SO_4$ molecules is negligible, the change





of particle mass is considered as the loss of $H_2SO_4$ mass from the gas phase to the particles. The
resulting change of saturation ratio per time unit is given as:
$$\frac{dS}{dt} = -\frac{R}{R_a}\frac{p}{p_{sat}}N_{ice}\frac{dm}{dt} \quad (B-2),$$

with the latent heat of vaporisation which is assumed as constant:
$$L = \frac{67.59 \cdot 10^3 \,\text{J mol}^{-1}}{M_{\text{H2SO4}}} \quad (B-3),$$

and $N_{ice}$ constitutes the number density of ice particles. Here, the sulphuric acid's molar mass is
$M_{\text{H2SO4}}$ = 0.098078 kg mol$^{-1}$. Note, the combination of the equations B$-$2 and B$-$1 depicts that
$\frac{dS}{dt} \sim r \cdot N_{ice}$, i.e. the temporal change of the precursor's saturation ratio is proportional to the
integral radius *IR* considered in Section 5.2.
In Figure B- 1 the variability of two aspects is considered and in the panels (a-c) it is
distinguished between three ice particle radii (1 μm, 10 μm, and 100 μm) and two different ice
particle number concentrations (0.01 and 0.1 cm$^{-3}$). The study by Ueyama et al. (2020) revealed
that ice particles (effective radii of about 15 μm) persist over 12 to 20 hours at convective
outflow levels between 365 K and 370 K potential temperature in the 2017 AMA.
Based on the simulation, apparently, the largest particles ($r_p$ = 100 μm) are capable to efficiently
suppress the NPF process. Particles of this size and in highest concentrations of 0.1 cm$^{-3}$ cause
the saturation ratio to rapidly abate to saturation level (i.e. $S$ = 1) within 20-50 minutes. Even at
lower concentrations (0.01 cm$^{-3}$) of particles of 100 μm radius, the saturation ratio is efficiently
reduced by more than 70 % within 1 hour. Particles of 10 μm radius and in concentrations of
0.1 cm$^{-3}$ appear to be equally efficient in reducing the saturation ratio by $\sim$ 70 % within 1 hour.
Smaller number concentrations of the same particle size range, and smaller particles ($r_p$ = 1 μm),
in general, require considerably more time than 1 hour to significantly reduce the $H_2SO_4$
saturation ratio.
In essence, cloud ice particles are well capable to rapidly reduce the saturation ratio of $H_2SO_4$
and, very likely, also the saturation ratio of other condensable gases. The ranges of $N_{ice}$ (0.01 -
0.1 cm$^{-3}$) and particle size (1 μm < $r_p$ < 100 μm) considered in the simulation correspond to the
characteristics of ice particles coincidently observed with NPF throughout the StatoClim 2017
mission (away from NPF higher concentrations and larger sizes were found, cf. Krämer et al.
(2020)). About 71% of all ice cloud detections in coincidence with NPF had an *IR* (i.e. $\overline{r_{ice}} \cdot N_{ice}$)
of less than 1 μm cm$^{-3}$, while only about 1.5 % of the ice particle samples reached *IR* values
greater than 7.5 μm cm$^{-3}$; the maximum *IR* of 24 μm cm$^{-3}$ was encountered once throughout the
entire mission. In general, the cirrus cloud particles are expected as coated with a $H_2SO_4/H_2O$





layer (Bogdan et al. (2006); Bogdan et al. (2013)) onto which sulphuric acid can condense.
However, impurities by weaker and substitutable acids (such as organic acids or HCl or HNO$_3$)
also allow the H$_2$SO$_4$ uptake on the surface, which could reduce the gaseous H$_2$SO$_4$ concentration
this way potentially suppressing NPF. Hence, in certain abundance the presence of cloud ice
particles restrains the NPF process, when condensation prevails over the competing gas-to-
particle conversion. The efficiency of condensation onto the ice particles' surface strongly
depends on

1)  the size and number concentration of cloud ice particles and,
2)  on the time interval during which the conditions remain at least saturated.

For the numerical simulation of the saturation decay, an ice particle is assumed as entirely
coated (consistent with Bogdan et al. (2006); Bogdan et al. (2013)) and the (real) ice particle's
habit (e.g. asphericity, porosity, etc.) remains unconsidered. Sporadic updraughts, such as
initialised by convective lifting well below the NPF level, or gravity waves could cause small-
scaled expansion and cooling which increases the precursor's supersaturation (Weigel et al.,
2020a). Hence, a certain concentration of H$_2$SO$_4$ molecules could exceed the supersaturation
threshold for NPF, even in the presence of abundant cloud ice, as long as the NPF process occurs
faster than the reduction of $S$ due to the present ice.

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





**Figure captions**
Figure 1: The flight patterns of the M-55 *Geophysica* during StratoClim 2017 over the Indian
subcontinent. New particle formation (NPF) encountered in clear air along the flight tracks are
indicated by orange colour in the main panel a). All NPF events coinciding with the detection of
cloud (ice) particles are coloured in blue. The general perspective, b), exhibits the patterns of the
eight StratoClim flights over Nepal, North - East India, Bangladesh, and the Bay of Bengal. For
more details, see Table 1.
Figure 2: The 1-Hz resolved number concentrations of aerosol particles in the ultrafine size
range ($N_{\mathrm{uf}}$) and of cloud (ice) particles ($N_{\mathrm{ice}}$) of the eight StratoClim flights compiled in one time
series ranging from 03:30 (UTC) to 12:30 (UTC). Kathmandu's (Nepal) local noontime is
indicated by the vertical orange line (corresponding to 06:15 UTC, or 22500 seconds of day,
UTC). Data points of $N_{\mathrm{uf}}$ in black whenever $N_{\mathrm{ice}}$ (cyan) equals zero, otherwise $N_{\mathrm{uf}}$ is coloured in
red. The blue dashed line indicates the median $N_{\mathrm{ice}}$ (0.031 cm$^{-3}$) for the entire dataset of cloud
particle detections during StratoClim 2017 (Krämer et al., 2020).
Figure 3: Vertical profiles of the mixing ratio (1-Hz resolved) of aerosols in the ultrafine size
range ($n_{\mathrm{uf}}$) versus the potential temperature ($\theta$). a): all data separated concerning coincident
detection of cloud (ice) particles (black: $N_{\mathrm{ice}}$ = 0 cm$^{-3}$, red: $N_{\mathrm{ice}}$ > 0 cm$^{-3}$) and b): all data coloured
correspondingly to the flight date, c): exclusively for $N_{\mathrm{ice}}$ > 0 cm$^{-3}$; d): when $N_{\mathrm{ice}}$ = 0 cm$^{-3}$. In the
panels at the bottom (e and f) in-cloud and clear-air measurements are distinguished
correspondingly to the intermediate panels (c and d) and coloured with reference to carbon
monoxide (CO) mixing ratios.
Figure 4: Histograms of the occurrence frequency of number concentrations $N_{\mathrm{uf}}$ of all NPF
detections (1-Hz resolved) throughout StratoClim 2017. a): all data of $N_{\mathrm{uf}}$ in general (black) and
separated concerning coincident detection of cloud (ice) particles in the diameter size range
3 μm < $d_{\mathrm{p}}$ < 937 μm (green: $N_{\mathrm{ice}}$ = $N_{\mathrm{3\text{-}937\mu m}}$ = 0 cm$^{-3}$, red: $N_{\mathrm{ice}}$ > 0 cm$^{-3}$). Hence, the sum of the green
and red curve yield the black curve, the vertical bars of which represent the square route of
count values. b): relative occurrence frequency of $N_{\mathrm{uf}}$ for in-cloud NPF (if detected coincidently
with $N_{\mathrm{ice}}$ > 0 cm$^{-3}$) normalised with respect to all NPF detections, i.e. the ratio of the absolute
occurrence frequencies (in red and black, Panel a). c): relative occurrence frequency of $N_{\mathrm{uf}}$ for in-
cloud NPF, if detected coincidently with various $N_{\mathrm{ice}}$ levels, which were normalised with respect
to those NPF detections with $N_{\mathrm{ice}}$ > 0 cm$^{-3}$, (in red, Panel b).
Figure 5: The total aerosol number concentration versus cloud particle number concentration in
accordance to de Reus et al. (2009). The total number concentration $N_{10}$ measured with one of
four COPAS channels together with coincident detections of $N_{\mathrm{ice}}$ (i.e. $N_{\mathrm{3\text{-}937\mu m}}$) by the NIXE-CAPS.
The data points are averaged over at least 10 s of flight time, and the bars exhibit the standard
deviation over the averaging periods. The data points are colour-coded in a) with reference to
*IWC*. b): NPF encounters (orange) throughout the averaging period (otherwise: grey). The blue-
shaded areas in both panels indicate the range of most of the data points provided by de Reus et
al. (2009).
Figure 6: NPF in the *IWC - T* parameter space (cf. Krämer et al. (2016)): measured ice water
content (*IWC*) coincidently detected with COPAS data as a function of ambient air temperature
throughout StratoClim 2017 (1-Hz resolved) – data points are colour-coded referring to (a) the





detected mixing ratios of ultrafine particles, $n_{uf}$, (b) the total mixing ratio $n_6$ measured with one
of four COPAS channels and (c) the carbon monoxide (CO) mixing ratio. Note: in (a), the data
points are grey if data of $n_{6-15}$ are available, while colours are apportioned only to those $n_{uf}$ (i.e.
$n_{6-15}$) complying with the NPF criterion. Generally, the black lines represent the median (solid)
and the upper-/lowermost bounds (dashed) of the core $IWC$ band, respectively, as obtained from
earlier measurements at other locations (Krämer et al. (2016)).
Figure 7: The 1-Hz resolved concentration of in-cloud detected ultrafine aerosol ($N_{uf}$) as a
function of the mean inter-crystal distance, $ICD$, between encountered cloud (ice) particles
colour-coded with reference to (a) the number concentration of cloud ice particles, (b) to $IWC$,
and (c) to the mean ice particles' radius.
Figure 8: The 1-Hz resolved concentration of ultrafine aerosol ($N_{uf}$) as a function of the cloud
(ice) particles' integral radius, $IR = \overline{r_{ice}} \cdot N_{ice}$ (with $\overline{r_{ice}}$, ice particles' mean mass radius) colour-
coded in correspondence to detected ice water content ($IWC$, panel a) and to measured CO
mixing ratio (b); in the absence of CO values the data points are blackened. The diagonal, grey-
coloured bars indicate a break-off edge along which the NPF seems limited by the $IR$ in general,
with two exceptional encounters of very recent or just proceeding NPF (see text for details).
Figure B- 1: Simulated change of the $H_2SO_4$ vapour's saturation ratio as a function of time due to
the presence of entirely $H_2SO_4$ - coated ice particle surfaces of various sizes and number
concentrations. a): particles with radii $r_p = 1\ \mu m$, b): $r_p = 10\ \mu m$; c): $r_p = 100\ \mu m$. Overall, this
simulation covers a range of integral radii $IR$ (= $\overline{r_{ice}} \cdot N_{ice}$) from 0.01 to 10 $\mu m\ cm^{-3}$. Note: a
cloud (ice) particle is assumed as coated with $H_2SO_4$ (consistent with Bogdan et al. (2006);
Bogdan et al. (2013)).

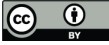



**Figures**

























Figure 1

























Figure 2




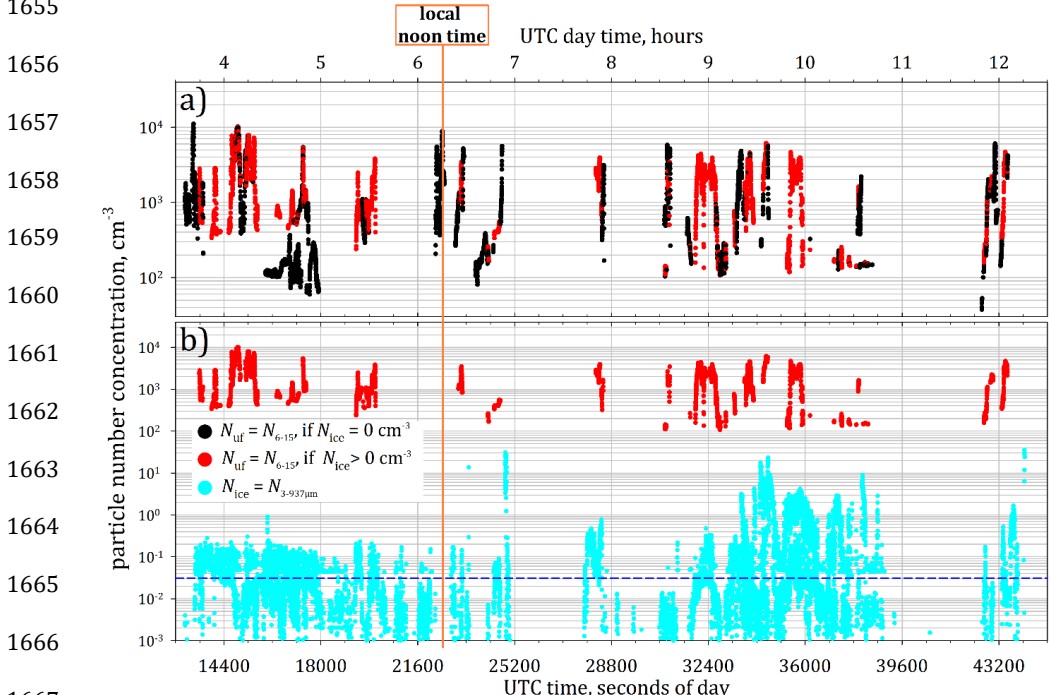

Figure 3


























Figure 4

































Figure 5








Figure 6


























Figure 7





Figure 8





Figure B- 1





**Table 1:** NPF data set of StratoClim 2017, separated by event detection under clear-air (i.e.
$N_{ice}$ = 0 cm$^{-3}$) and in-cloud conditions (i.e. $N_{ice}$ > 0 cm$^{-3}$). Discussed in-cloud NPF events (104
incidents that comply with introduced NPF criterion, Section 2.2) are partially embedded in
larger clear-air NPF fields with continuously elevated $N_{uf}$. The total number of measurement
seconds with NPF detections under either of both conditions is scaled to the total data set of the
CN measurements and the total duration of NPF encounters. The mean horizontal distance is
calculated from the event duration based on a mean flight speed of the M-55 *Geophysica*
(154 ± 39 m s$^{-1}$, variable flight attitude remains unconsidered) and may be understood as
equivalent horizontal extension of a NPF event. The total measurement time of in-cloud NPF
encounters is categorised into vertically stacked bins of 5 K potential temperature.

| NPF condition | total duration | | percentage of | | mean horizontal |
| | seconds | hh : mm | NPF data | total dataset | distance in km |
| --- | --- | --- | --- | --- | --- |
| clear-air | 4866 | 01 : 21 | ∼ 51.2 % | ∼ 5.3 % | ∼ 750 |
| in-cloud | 4634 | 01 : 17 | ∼ 48.8 % | ∼ 5.0 % | ∼ 714 |

| in-cloud NPF | | | | |
| --- | --- | --- | --- | --- |
| potential temperature | total duration | | percentage of in-cloud NPF | mean horizontal distance in km |
| | seconds | hh : mm | | |
| 355 – 360 K | 432 | 00 : 07 | ∼ 9.3 % | ∼ 67 |
| 360 – 365 K | 1231 | 00 : 21 | ∼ 26.6 % | ∼ 190 |
| 365 – 370 K | 1455 | 00 : 24 | ∼ 31.4 % | ∼ 224 |
| 370 – 375 K | 1375 | 00 : 23 | ∼ 29.7 % | ∼ 212 |
| > 375 K | 141 | 00 : 02 | ∼ 3 % | ∼ 22 |
