# Peer review of "New particle formation inside ice clouds: In-situ observations in the tropical tropopause layer of the 2017 Asian Monsoon Anticyclone"

_Atmospheric Chemistry and Physics, 2020_

## Author Response (AR1)

**Authors' replies to the reviews of the ACP manuscript** acp-2020-1285

"New particle formation inside ice clouds: In-situ observations in the tropical tropopause layer of the 2017 Asian Monsoon Anticyclone"

Please note the changed title: "In-Situ observation of New Particle Formation (NPF) in the tropical tropopause layer of the 2017 Asian Monsoon Anticyclone: Part II - NPF inside ice clouds"

On behalf of all authors, I would like to express our appreciation to the two reviewers for their valuable and constructive suggestions. We are very grateful, as these have helped to improve and complete the present study. We hope to have addressed all comments adequately and hereby submit a revised and much shortened version of the article for re-evaluation and thank the reviewers in advance for their renewed efforts.

Anonymous Reviewer #1 (Rev1)

Review

Where the manuscript falls short is the valiant, but lengthy, complex, and mostly inconclusive attempt to produce quantitative relationships between ice cloud properties and ultrafine aerosol concentrations / NPF strength (Sec. 5). Relationship that are identified are weak and only applicable to subsets of the collected data, producing much hypothesizing and conditional statements that prevent strong conclusions. The main insight is that below the integrated radius (IR) threshold of ~ 1 µm cm-3, the observed ultrafine aerosol concentrations are independent of IR, while above it, the maximum concentration of ultrafine aerosol falls linearly with increasing IR. This insight may hold in general, but it also could be limited to this data set. The reason why it is so difficult to construct quantitative relationships between the observed quantities is, in this reviewer's opinion, the complexity and nonlinearity of the processes (transport, precursors, scavenging, mixing, chemical conversion, time since NPF, etc.) that shape the observations, rather than any failure on the part of the authors.

This reviewer's recommendation is to focus on the very significant results and insights in this work, of which there are plenty, and drop the complex, hypothetical, conditional, and inconclusive elements of the analysis that weigh the manuscript down and compromise the overall quality of the work. The manuscript would also benefit from proofreading for English; some expressions are used in an unusual way ("constrained" is a better expression than "confined" in many places, etc.), and the language could be simplified for clarity and ease of reading. A major revision is recommended to provide sufficient time for any necessary changes.

> [Authors]: We see the point the reviewer is making here and gratefully acknowledge that the reviewer has perceived the link between the NPF limitation through the *IR* as a key outcome of the work. Of course, the connections show results from the data of the StratoClim campaign only. Further studies will have to show whether the results are transferable to the general validity. In the revised version, we hope to have made clearer
>
> a) the importance of the *IR*,
> b) that the relationship between NPF and *IWC* is less conclusive albeit *IWC* and *IR* are proportional, and
> c)  that the number or size of ice particles alone is not the limiting factor for the NPF rate.

> With this article, we hope to refine the findings from previous work (e.g. Weigel et al. 2011) on this dependence of NPF and cloud ice with some more in-depth and comprehensive specification.

Specifics

[Rev1]: Line 58: "From the CLOUD experiments, which were performed under a variety of controlled conditions, it can be deduced that the intensity of NPF (the formation rate of new particles per air volume and per time unit) depends on the concentration of the NPF precursors." Hasn't this been known long before the CLOUD experiments - e.g., https://doi.org/10.1029/2003JD004460, and others?

> [Authors]: The passage has been rephrased.

[Rev1]: Line 65: ... indeterminable, because ... (expand a little bit on this for the reader's sake)

> [Authors]: The sentence is more clearly formulated in the revised manuscript. Please refer to this point in the new paper version.

[Rev1]: Line 76: "however" may not be necessary here - could be dropped for simplicity.

> [Authors]: corrected as suggested.

[Rev1]: Line 77: ". Under real conditions in the atmosphere, however, the concentration of precursor material is spatially and temporally highly variable."

Pleas provide one or two references.

> [Authors]: done as suggested.

[Rev1]: Line 93: "Investigations concerning the occurrence of NPF within clouds, or in their immediate vicinity, are sparse ..."

This statement would appear incompatible with the work of Clarke and Kapustin (2006), who report decade of data on particle production, transport, evolution, and mixing in the troposphere, much, if not most of it, near clouds.

> [Authors]: The sentence was misleadingly worded and in the revised version of the article, this passage was rephrased.

[Rev1]:  Line 95: "... possible reasons for this are discussed by Wehner et al. (2015)."

Please briefly give some of these reasons - this will be illuminating to the reader.

> [Authors]: In the revised manuscript two of the main reasons are provided, which were discussed by specified authors.

[Rev1]: Line 229 "... principally based on the difference of both quantities (cf. Weigel et al. (2011)). "  This could be removed.

> [Authors]:removed as suggested

[Rev1]: Line 247: "coincidently"    replace with "coincide"

> [Authors]:corrected as suggested

[Rev1]: Line 280: "computations"     replace with "computational"

> [Authors]:corrected as suggested

[Rev1]: Line 375: "The encountered in-cloud NPF events at altitudes between approximately 11 km and 16.5 km (~ 355 ? 385 K) had a mean event duration of 14.5 seconds (ranging from one second to a maximum of about 300 seconds)."

"event duration" means "flight time spent in air with in-cloud NPF", is that correct? If yes, please make sure that this is clear, because the reader might otherwise assume that this refers to the time period during which NPF took place.

> [Authors]: The revised paper version implies corresponding definition in the Section 2.2 Terminology and notations.

[Rev1]: Line 648: " It is not likely that a high number of interstitial, non-activated aerosol is accountable for the abundance of submicrometre-sized particles."

Please substantiate that "it is not likely", or if substantiation is not possible, remove the passage.

> [Authors]: The sentence was rephrased in the revised article version

[Rev1]: Line 666: "... likely suffice ..."  Please substantiate this, or if substantiation is not possible, remove the passage.

> [Authors]: the questionable formulation was revised.

[Rev1]: Line 912: "The IR turned out as appropriate cloud ice related parameter to juxtapose with NPF data." This is a very confident statement given the very limited explanatory power of the IR.

> [Authors]: As already indicated above, we have more clearly emphasised the role of the *IR* as the control variable in the mass increase on particles' growth. Based on the direct proportionality between the integral radius and the particle weight increase, the proportionality between *IR* and the change in the saturation ratio (Appendix B) can immediately be derived. The fact that the limitation of NPF by *IR* is so visible in the data strengthens the hypothesis concerning the main process, which limits in-cloud NPF: the precursors deplete on the ice crystal surfaces, whereas the growth of the ice crystals is proportional to *IR*, hence, understandably the concentration of the precursors depends on the *IR*. Since the *IR* is a commonly used variable in the modelling of the growth process of ice particles (*IR* directly enters the numerical equations describing the particle growth), we would refrain from questioning the validity of the *IR* as suggested by the referee.

Anonymous Reviewer #2 (Rev2)

**[...]** The paper has a bit too much text devoted to qualitative and/or speculative details, and a long-winded writing style which sometimes detracts from its key messages. However, if my comments below can be addressed, this generally very good paper will be well worthy of publication in ACP and it should be highly cited.

**Major comments:**

[Rev2]: Based on the scatter in Figure 8, the integral radius doesn't work as well for controlling NPF as one might hope. Of course, given the difficulties of measuring NPF on an aircraft, not knowing accurately precursor concentrations or air mass history, one should not expect too

much. Perhaps because of this, the authors don't currently present quantitative metrics for whether or not the IR is any use.

So maybe the authors can disentangle the data more brutally to extract some numerical information on the usefulness of the IR? If they excluded data with IWC below a threshold, say 0 on their log scale in Figure 8, then is there some correlation coefficient between IR and N_uf? Even if no meaningful correlation can be presented, perhaps the message can be firmed up with further stratification of the data? Otherwise, it is hard to justify the lengthy text and detailed discussion associated with the IR in the paper, and the authors could instead substantially streamline these sections and perhaps focus instead on drawing out more quantitative conclusions about the role of in-situ vs liquid-origin cirrus.

> [Authors]: We specified the role of the *IR* more clearly in the revised paper version. The *IR* is the control variable for mass increase on particle growth. Based on the direct proportionality between the integral radius and the particle mass increase, the proportionality between *IR* and the change in the saturation ratio (Appendix B) is directly derivable. As the limitation of the NPF by IR is so clearly visible in the scatter plots strengthens the hypothesis regarding the main process, which limits the in-cloud NPF. The NPF precursors deplete due to their uptake by the ice particles as a function of *IR*, which is a commonly used variable in the modelling of the growth of ice crystals that enters directly into numerical equations, such as the equation describing particle growth.
>
> Regarding the referee's further remarks in this context: A derivation of a functional relationship between IR and $N_{nm}$ is not straightforward, since, as Referee 1 has also noted, this relationship is possibly a special feature in the UT/LS of the AMA. Moreover, the grey markings in the figure are by no means to be understood as sharp boundaries. As was shown by a couple of data points from two independent measurement flights, there will also be data points beyond the grey markings. However, the probability of observation decreases as the grey markings are crossed. The revised version of the article has been shortened considerably and also streamlined at this point according to the reviewer's suggestions.

Weigel et al 2020a is only referred to for specific details, why not take advantage of it for the broader context to avoid repetition (e.g of lines 173-193 in Weigel et al 2020a), and maybe even call this Part II of a 2-paper series (something to discuss with the editors)? The current situation is quite confusing. As another example, the first paragraph of the summary is background information that describes findings that are very similar to those in Weigel et al 2020a and there was not sufficient referencing provided.

> [Authors]: In agreement with several reviewer comments on the two StratoClim campaign NPF papers, which were recently discussed at ACPD:
>
> * in the revision of this manuscript we have also deleted essential parts of the text that can be counted as basic knowledge.
>
> * deleted some dispensable explanations and instead referred to the introduction in the companion paper.
>
> * the reviewer's suggestion to change the title was thankfully adopted for both papers.

**Minor comments**

[Rev2]: The diversity of units used in the field is confusing (not the authors' fault). Will be helpful to present aerosol number concentrations in cm-3 alongside mg-1 (as concentration in cm-3 is relevant for the molecular collision frequency leading to NPF) and altitudes in km alongside K and hPa throughout the text and especially in section 3.2.

> [Authors]: In Section 3.2, particularly in the conclusions Section, and where else the observations were generally summarised, the specifications of particle number concentrations and geometric heights were included in the revised version of the manuscript. Also Table 1 was expanded by a column with geometric heights. It is understandable that the use of special units is advantageous for different applications. In the present work, the aim was to make the measurements comparable - across the StratoClim campaign observations, but also with earlier measurements from other regions and other seasons for which mass mixing ratios and potential temperature are well suited.

[Rev2]: Figure 2 might be more helpful as a frequency distribution of number concentrations exceeding a threshold, or if the existing figure is accompanied by something like that.

> [Authors]: The reviewer makes an important point here and we agree on the matter. Figure 2 has been updated and the distribution of incident counts has been supplemented for different threshold values of the number concentration $N_{nm}$. The caption and text have been amended accordingly. Please refer to the revised manuscript version.

[Rev2]: L389: "Furthermore, there is no obvious indication that the number of ice particles present had a direct influence on the NPF strength" seems inconsistent with later L516 "Although an ultimate observational evidence is currently lacking, however, these findings suggest that NPF is entirely prevented in cases when Nice substantially exceeds 2-3cm-3" – maybe add a qualification to the earlier statement to make this later statement seem less at odds with it.

> [Authors]: We follow the reviewer's argument and have deleted the first sentence in the revised manuscript version.

[Rev2]: L409 Could cite and discuss Bianchi et al 2020 (an understandable omission given the date this article was posted). https://www.nature.com/articles/s41561-020-00661-5

> [Authors]: We fully agree with the referee that the articles mentioned should be cited. However, we find the passage indicated here to be less appropriate, since the results in Bianchi at al 2021 come from ground-based measurements at a high mountains site and the effect in the free troposphere is inferred as follows: "…which suggests that the whole Himalayan region may act as an 'aerosol factory' and contribute substantially to the free tropospheric aerosol population." The lifetime of freshly formed particles is limited to hours (Weigel et al. 2020a). It would be difficult to explain how the near-surface NPF relates to the observations in the UT/LS without providing speculations on very short transport. Nevertheless, the article has now been referenced elsewhere in submitted and revised article as well as in the revised version of Weigel 2020a.

[Rev2]: L429 what about ammonia?

> [Authors]: Indeed, here the referee makes an important point and a statement about the retention of NH4 is added in the revised version of the article.

[Rev2]: Personally, I find it much easier to read and review papers, especially long ones, if the figure captions are on the same page as the figures (and preferably presented when they are first mentioned in the text rather than at the end, though this is less important). By the time I have found the figure that relates to a point in the text, opened the paper in two more instances of my browser, found its caption, looked back to the figure, understood the caption, gone back to the figure, understood the figure, I have forgotten why I was interested. Maybe this is a matter of opinion, but the ACP guidelines here: https://www.atmospheric-chemistry-and-physics.net/submission.html#reviewfiles say "Figures and tables as well as their captions must be inserted in the main text near the location of the first mention (not appended to the end of the manuscript)" so it seems I am not alone.

> [Authors]: We regret the circumstances and the fact that the Copernicus guidelines have been interpreted so liberally. The subdivision is dedicated to the old school, where the text, illustrations and captions were printed on paper and edited in three stacks. In future, I will dispense with this subdivision. In the final set, the layout of the images, tables and captions in the running text will be optimised by Copernicus.

[Rev2]: While generally nicely written, the paper is long, and the writing could often be more economical. I encourage the authors to go through each paragraph sentence by sentence as if there was a page limit, and use more efficient phrasing and omit unnecessary details. The paper would be easier to read and the authors would save on page charges.

> [Authors]: The revised version of the article was edited and shortened according to the reviewer's suggestion.

[Rev2]: A few sentences are written confusingly: "In particular, the abundance of in-cloud NPF concentrates between ratios of 1:30000 and 1:500000, which may not further surprise, as the large aerosol number concentrations are indicative to result from NPF." What is an "NPF concentrate"?

> [Authors]: In the course of the revision, complicated sentence structures were broken up and restructured to improve the readability.

[Rev2]: Also, there are some typographical errors; I pick out only examples. Many commas (e.g. before "that") reminiscent of German should be removed. On line 234 "principle" is confused with "principal". Finally, while not strictly incorrect, manuscripts are usually "drafted", not "draughted". See https://www.merriam-webster.com/words-at-play/usingdraft-and-draught

> [Authors]: In revising the manuscript, the commas were carefully checked, the word "principle" was deleted without replacement, and the expression "drafted manuscript" was corrected according to the reviewer's suggestion.